# Cellular and Molecular Targets for Non-Invasive, Non-Pharmacological Therapeutic/Rehabilitative Interventions in Acute Ischemic Stroke

**DOI:** 10.3390/ijms23020907

**Published:** 2022-01-14

**Authors:** Gelu Onose, Aurelian Anghelescu, Dan Blendea, Vlad Ciobanu, Cristina Daia, Florentina Carmen Firan, Mihaela Oprea, Aura Spinu, Cristina Popescu, Anca Ionescu, Ștefan Busnatu, Constantin Munteanu

**Affiliations:** 1Faculty of Medicine, University of Medicine and Pharmacy “Carol Davila”, 020022 Bucharest, Romania; cristina.daia@umfcd.ro (C.D.); mihaelamandu37@yahoo.com (M.O.); aura.spinu@umfcd.ro (A.S.); anca.ionescu@umfcd.ro (A.I.); stefan.busnatu@umfcd.ro (Ș.B.); 2Neuromuscular Rehabilitation Clinic Division, Teaching Emergency Hospital” Bagdasar-Arseni”, 041915 Bucharest, Romania; aurelian.anghelescu@umfcd.ro (A.A.); cristina_popescu_recuperare@yahoo.com (C.P.); 3Faculty of Midwives and Nursing, University of Medicine and Pharmacy “Carol Davila”, 020022 Bucharest, Romania; 4Faculty of Medicine, University ”Titu Maiorescu”, 0400511 Bucharest, Romania; danblendea@gmail.com; 5Physical and Rehabilitation Medicine & Balneology Clinic Division, Teaching Emergency Hospital of the Ilfov County, 022113 Bucharest, Romania; firancarmen@yahoo.com; 6Computer Science Department, Politehnica University of Bucharest, 060042 Bucharest, Romania; vlad.ciobanu@upb.ro; 7Faculty of Medical Bioengineering, University of Medicine and Pharmacy” Grigore T. Popa”, 700115 Iași, Romania

**Keywords:** systematic review, cellular and molecular targets, non-invasive, non-pharmacological interventions, neurorestorative, neuroregeneration, brain repair

## Abstract

BACKGROUND: Cerebral circulation delivers the blood flow to the brain through a dedicated network of sanguine vessels. A healthy human brain can regulate cerebral blood flow (CBF) according to any physiological or pathological challenges. The brain is protected by its self-regulatory mechanisms, which are dependent on neuronal and support cellular populations, including endothelial ones, as well as metabolic, and even myogenic factors. OBJECTIVES: Accumulating data suggest that “non-pharmacological” approaches might provide new opportunities for stroke therapy, such as electro-/acupuncture, hyperbaric oxygen therapy, hypothermia/cooling, photobiomodulation, therapeutic gases, transcranial direct current stimulations, or transcranial magnetic stimulations. We reviewed the recent data on the mechanisms and clinical implications of these non-pharmaceutical treatments. METHODS: To present the state-of-the-art for currently available non-invasive, non-pharmacological-related interventions in acute ischemic stroke, we accomplished this synthetic and systematic literature review based on the Preferred Reporting Items for Systematic Principles Reviews and Meta-Analyses (PRISMA). RESULTS: The initial number of obtained articles was 313. After fulfilling the five steps in the filtering/selection methodology, 54 fully eligible papers were selected for synthetic review. We enhanced our documentation with other bibliographic resources connected to our subject, identified in the literature within a non-standardized search, to fill the knowledge gaps. Fifteen clinical trials were also identified. DISCUSSION: Non-invasive, non-pharmacological therapeutic/rehabilitative interventions for acute ischemic stroke are mainly holistic therapies. Therefore, most of them are not yet routinely used in clinical practice, despite some possible beneficial effects, which have yet to be supplementarily proven in more related studies. Moreover, few of the identified clinical trials are already completed and most do not have final results. CONCLUSIONS: This review synthesizes the current findings on acute ischemic stroke therapeutic/rehabilitative interventions, described as non-invasive and non-pharmacological.

## 1. Introduction

Stroke is a significant cause of death or long-term disability, with a severe negative impact on the affected individuals, their kin, and society, and a global incidence of one in every six people. As estimated by the World Health Organization, 15 million people suffer stroke worldwide each year [1]. Stroke has two subcategories: hemorrhagic and ischemic (IS). The ischemic subgroup is responsible for about 87% of all strokes [2]. The acute time framing for both types of stroke refers primarily to a time window of the first few days after the onset of such a pathologic event [3,4].

Neurovascular dysfunction [5] is a life-threatening condition [6]. The ischemic stroke is a transient or permanent interruption of the blood supply into/through the cerebral vasculature. During ischemia, lack of oxygen and energy supply generates inflammation and oxidative damage. Even if essential after the acute ischemic episode, reperfusion can also induce secondary injuries. Moreover, it has been reported that reperfusion after prolonged ischemia had more harmful effects than the ischemia itself [7]. For instance, acute stroke patients subject to reperfusion intervention with recombinant tissue plasminogen activator (rtPA) are at greater risk of developing seizures than non-treated patients [8].

In approximately one-third of ischemic stroke patients, embolism to the brain originates from the heart, especially in atrial fibrillation. Extracranial artery stenoses are prone to destabilization and plaque rupture, leading to cerebral thromboembolism [9]. Additionally, ischemic stroke can be caused by in situ minor vessel diseases or arterial embolism. Thromboembolic occlusion of significant or multiple smaller intracerebral arteries leads to focal impairment of the downstream blood flow and secondary thrombus formation within the cerebral microvasculature [10]. The brain area with severely impaired blood flow is the “infarct core,” representing the epicenter of a stroke, and the neighboring area is named the “ischemic penumbra”, a salvageable zone. Ischemia-related pathologic consequences can last for days or even weeks in these brain territories. In the penumbra, less severe damage occurs depending on numerous factors, including the activation of specific genes, which ultimately may result in apoptosis [11].

The human brain receives close to 15% of the body’s resting cardiac output blood flow while expending 20% of its oxygen. As the brain is highly susceptible to hypoxia or ischemia, protecting brain tissue from injury, simplified as “neuroprotection”, has been a long-sought-after strategy when quelling physiological damage following stroke onset.

The energy requirements of the brain are relatively high compared to other organs. Therefore, ischemic brain injury initially leads to the excessive consumption of high-energy phosphates, particularly adenosine triphosphate (ATP) and phosphocreatine [12]. As a result, many biochemical and physiological changes occur due to the interruption of blood flow to the brain [13], including energy depletion [14], anaerobic production of lactate [8], compression of brain capillaries [8], and reduced blood reflow due to the swelling astrocytes [15]. In addition, the electrolytic balance is disturbed with the fall in ATP level, which results in potassium leakage from the intracellular compartment and an influx of sodium and calcium along with the remarkable increase in cellular water content due to Na^+^/K^+^ATPase and voltage-gated calcium channel dysfunctions. These are interrelated and coordinated events, leading to ischemic necrosis in the severely affected ischemic-core regions. Metabolically active zones surround the necrotic core regions, the ischemic penumbra, which comprises about half of the total lesion volume and represents the region in which there is an opportunity for post-stroke therapy [5].

Decades of research efforts have investigated over 1000 pharmacological variants, including excitatory amino acid antagonists, calcium channel blockers, free radical scavengers, and growth factors thought to target and correct different pathophysiological conditions. Unfortunately, almost all attempts have failed to make a successful transition into clinical use. Intravenous rtPA is the only effective treatment in acute stroke [16].

Non-pharmaceutical therapies are seen as promising strategies for stroke. Indeed, accumulating data suggest that “non-drug” approaches might provide new opportunities for stroke therapy, such as therapeutic hypothermia, acupuncture, certain medical gases, and other strategies [17].

A few clinical trials have been found regarding non-invasive, non-pharmaceutical interventions in acute ischemic stroke. Instead, animal studies were used to obtain information on recovery mechanisms after stroke. An experimental stroke alters the expression of many genes, leading to increased levels of key growth factors, increased synapses and dendrites, axonal remodeling and angiogenesis, and increased brain excitability, mediated by changes in glutamate gamma-aminobutyric receptor subtypes. (GABA). These events are often concentrated in, but are not limited to, the perilesional tissue [18].

Physiological responses to stroke can be organized into three broad temporal phases. The first phase includes the first hours after the stroke onset and can save the threatened tissue, for example, by reperfusion or neuroprotection. The second phase begins from a few days to weeks after the stroke and corresponds to the peak weeks of spontaneous neuronal repair. Finally, the third phase is a chronic one in which the brain is relatively stable in terms of endogenous events related to repairs, but changes in brain structure and function are still possible with specific interventions. These three phases delimit distinct biological states and have clinical implications for the administration of restorative therapies [19].

Recovery after stroke has been explained as a rich cascade of events encompassing cellular, molecular, genetic, demographic, and behavioral components. These factors have been proven as covariates in therapeutic trials of restorative agents with a sound neurobiological basis. In addition, advances in functional neuroimaging and brain mapping methods have provided a valuable parallel data collection system for stroke recovery in humans.

The observation that both protective and pathological cascades are coactivated by ischemia suggests that the potentiation of protective signaling pathways may block the cytotoxic effects of ischemia; however, how to potentiate the neuroprotective program that is characteristic of ischemic responses remains unclear [20].

Achieving a systematic review and meta-analysis is, and continues to become, increasingly elaborated and rigorous. The current methods include the recently updated PRISMA methodology, including registration on specific online platforms such as PROSPERO. Therefore, we consider this review to contribute to the understanding of the main biological and pathological processes targeted by non-invasive, non-pharmacological therapeutic/rehabilitative interventions in acute ischemic stroke, as synthetically presented in Figure 1. To our knowledge, such endeavors are scarce in the manner proposed by the new PRISMA statement. In addition, adding some monitoring-related data to non-invasive, non-pharmacological therapeutic/rehabilitative interventions will hopefully provide a broader spectrum of evidence and, consequently, a better understanding of this complex yet insufficiently solved pathology: acute ischemic stroke.

## 2. Materials and Methods

To present the state-of-the-art regarding the main currently available, non-invasive, non-pharmacological related interventions in acute ischemic stroke, we have accomplished a systematic and synthetic literature review based on the PRISMA guidelines. Accordingly, we have interrogated, employing specific keyword combinations (see Appendix A), five international medical databases: Elsevier, National Center for Biotechnology Information (NCBI)/PubMed, NCBI/PubMed Central (PMC), Physiotherapy Evidence Database (PEDro), and—to verify whether the works found are published in ISI-indexed journals—the renowned ISI Web of Science database.

For our PRISMA-type method, an adapted flow diagram of bibliographic resources search and filter/selection, we considered only free, full-text-available papers, written in English, which appeared between 01 January 2018 and 31 December 2020. After this first step, we removed duplicates (same work found in two or more databases); in the third step, we checked for and retained only those issued in ISI-indexed publications. Further, in the fourth step, we indirectly evaluated the scientific impact/quality of each of the remaining articles, using our own, customised, quantification-weighted algorithm—PEDro classification/scoring-inspired—considering works that obtained a score of at least 4 (“fair quality = PEDro score 4–5”); in the fifth step, we conducted a direct qualitative analysis of the 66 selected articles and eliminated those that seemed eligible according to the above mentioned criteria, but, after analysing their full texts (“full-text articles excluded, with reasons”), we determined to not contain information with a consistent connection to our subject. Consequently, we also made a final qualitative and quantitative selection: ultimately, we kept and used information from 54 papers (see Figure 2, showing our completed, adapted PRISMA-type of flow diagram [21] and Appendix A, with authors, titles, journals and related links to the works that were ultimately selected in our systematic literature review, included in the Appendix A). It should be mentioned that, although we thoroughly fulfilled the PRISMA-type method of search and selection for this systematic literature review, some papers of interest could still be missed. We believe that appropriate, freely found, bibliographic resources, if added to those acquired/selected according to the above-described standardized method, could be—and proved to be—helpful to enhance knowledge of the subject.

To evaluate non-invasive, non-pharmacological therapeutic/rehabilitative interventions in acute ischemic stroke, we searched three well-known platforms: https://clinicaltrials.gov (accessed on 16 December 2021), https://trialsearch.who.int/ (accessed on 16 December 2021) and https://www.clinicaltrialsregister.eu/ (accessed on 16 December 2021) for clinical trials using each of the following search items and looking for synthetic reviews: electro-/acupuncture, hyperbaric oxygen therapy, hypothermia/cooling, photobiomodulation, therapeutic gases, transcranial direct current stimulations, or transcranial magnetic stimulation. In addition, the inclusion criteria were fixed regarding the characteristics of the intervention: non-invasive and non-pharmacological, patients with acute ischemic stroke (as diagnosed using any recognized diagnostic criteria), age 18 to elderly, of all genders. Exclusion criteria corresponded to a patient age of less than 18 years. All the aforementioned literature and clinical trial databases were searched in November and December 2021.

The inclusion of articles in the synthesis was first based on a PEDro-adapted quantitative method, considering the number of citations per year. All the values were limited to between 0 and 10 using a mathematical equation. For the qualitative selection, three specific items were used: the studies describe non-invasive, non-pharmacological interventions, the studies describe cellular/molecular mechanisms for non-invasive interventions, and present positive effects/negative effects for non-invasive interventions, each scored with 0 if absent and 10 if present. The same mathematical equation was applied to limit the results to between 0 and 10. Only articles which reached at least 4 points were selected.

A meta-analysis was included to analyze the abovementioned interventions’ frequency in the acute ischemic stroke clinical trials.

## 3. Results

The initial number of obtained articles was 313. After fulfilling the five steps of the filtering/selection methodology, 54 fully eligible papers were selected for synthetic review. We enhanced our documentation with other bibliographic resources with an overall connection to our subject, identified in the literature with a non-standardized search, to fill the knowledge gaps. Fifteen clinical trials were also identified.

### 3.1. Cellular and Molecular Mechanisms as Presumptive Therapeutic/Rehabilitative Targets in Acute Ischemic Stroke

Cellular and Molecular Mechanisms as Presumptive Therapeutic/Rehabilitative Targets in Acute Ischemic Stroke (Figure 3).

#### 3.1.1. Transcriptional Factors

Epigenetics has gained much attention due to increasing evidence suggesting its role in the development of many diseases, as well as being used to explain therapeutic interventions, including in ischemic stroke [22]. In addition, members of the *signal transducers and activators of the transcription* (STAT) family, *nuclear factor kappa-light-chain-enhancer of activated B cells* (NF-κB), *nuclear factor erythroid 2- related factor 2* (Nrf2), and *peroxisome proliferator-activated receptors* (PPARγ) are essential transcription factors associated with various molecular mechanisms that are also triggered in the acute ischemic stroke.
*STAT family members,* STAT1-STAT6, found in the cytoplasm, play an essential role in regulating inflammatory responses. *Janus kinases* (JAKs) and STATs pathways handle dozens of cellular responses such as proliferation, differentiation, migration, apoptosis, and cell survival, depending on the cellular context. The *JAK2/STAT3* pathway has been highlighted by in vitro and in vivo experimental stroke models, subsequently activating numerous genes responsible for many cellular functions in neural injury and repair [23].*NF-κB* activates the microglia and transforms these activated cells into the M1 phenotype. It is activated in response to detrimental stresses and induces inflammatory responses. The activated NF-κB is translocated to the nucleus, promoting the secretion of pro-inflammatory cytokines, such as IL18, IL6, and TNF-α [24].*Nrf2* is a redox-sensitive transcription factor that demonstrates both antioxidant and anti-inflammatory properties, and has been identified as having a protective effect following ischemic stroke [24].*PPARγ* is a ligand-activated transcription factor. In acute ischemic stroke, PPAR γ is also activated and has been shown to reduce tissue damage by directly inhibiting the NF-κB pathway, reducing inflammation, and stimulating the antioxidant response element (Nrf2/ARE) axis to decrease oxidative stress [24].*Non-coding RNAs* (ncRNAs) are a class of functional RNAs that regulate post-transcriptional gene expression [25]. For example, *H19* is a long ncRNA (lncRNA) that induces the onset of ischemic stroke and participates in the chronic regeneration process following ischemic stroke. In addition, *H19* has been shown to promote M1 microglia polarisation and induce neuroinflammation in ischemic stroke by regulating histone deacetylases. In clinical experiments, *H19* levels were identified to be significantly increased in the plasma at 3 h, 7, 30, and 90 days post-ischemic stroke [25].Micro RNAs (miRNAs) are small ncRNAs [26] that mediate post-transcriptional gene regulation by controlling the translation of mRNA into protein. In ischemic stroke, miRNAs are involved in multiple cellular functions, such as neuronal development, injured tissue repair, remodeling, and different neuronal activities, and their target genes play a crucial regulatory role in the inflammatory process of post-ischemic reperfusion injury, which explains their potential use as a therapeutic target in ischemic stroke [24].

#### 3.1.2. Receptors

Membrane receptors such as *toll-like receptor 4* (TLR4), *sphingosine 1 phosphate receptors* (S1PRs), and *thromboxane A2 receptor* (TXA2R) are closely associated with microglia polarization, seen in ischemic stroke.
*TLR4, predominantly expressed by microglia, is an essential regulator of inflammatory responses. Elevated TLR4 protein levels are associated with poor outcomes in patients with ischemic stroke. Furthermore, in ischemic stroke, TLR 4 has been shown to recognize molecules related to damage, such as lipopolysaccharides (LPS) [27].**S1PRs: S1PR1-S1PR5 are G-protein-coupled receptors expressed in abundance in the microglia and are demonstrated to regulate inflammatory responses following ischemic stroke. S1P is the ligand protein for S1PR and has been identified to bind to S1PR1, S1PR2, and S1PR3 to trigger neuroinflammatory reactions in ischemic stroke [24].**TXA2R, another G-protein-coupled receptor, can promote platelet activation and aggregation by regulating thrombosis/hemostasis and inflammatory responses. In ischemia/reperfusion, significant levels of TXA2R expression were detected in microglia/macrophages [24].*

#### 3.1.3. Ion and Ionic Channel Proteins

The expression of ion channels is changed in response to the voltage and pH gradients in the microenvironment [24]. In ischemic stroke, previous studies have identified that changes also occur in the expression levels of K^+^ channels, which can regulate microglia polarization. K^+^ channels are membrane proteins involved in the rapid and selective flow of K + ions across the cell membrane, generating electrical signals in cells. [28].

Ca^2+^-activated potassium channels (K_Ca_ channels) represent a heterogeneous family of ion channels that play a neuroprotective role by counteracting NMDA-induced neurotoxicity. K_Ca_ channels are expressed at inner mitochondrial membranes (mitoK_Ca_), where they have been suggested to be involved in neuroprotection by different mechanisms [29].

Emerging evidence has recently shown that transient receptor potential melastatin 2 (TRPM2), the most frequent transient receptor potential (TRP) channel, is a Ca^2+^-permeable, nonselective cation channel. TRPM2 is highly expressed in the central nervous system and is activated by hydrogen peroxide (H_2_O_2_) and agents that produce reactive oxygen/nitrogen species, increasing the Ca^2+^ concentration. In addition, numerous studies have revealed that TRPM2 is detrimental to brain ischemia. Reactive oxygen species (ROS)-induced, TRPM2-dependent, delayed neuronal cell death may represent a common mechanism in ischemic stroke. TRPM2 activation results in free oxygen radical production caused by depolarization of the mitochondrial membrane, the influx of Ca^2+^, the release of apoptotic factors (including caspases 3 and 9), and the eventual cell death in the neuronal lesion of the cerebral hippocampus induced by cerebral ischemia. These results support a key role for TRPM2 in coupling PKC/NOX-mediated ROS generation, which causes subsequent positive feedback loops for ROS-induced delayed cell death. [30].

Ischemic, stroke-induced TRPM2 activation leads to significantly increased extracellular zinc ions. Consequently, Zn^2+^-induced neuronal death involves lysosomal and mitochondrial dysfunction. Additionally, the TRPM2 channel’s genetic deletion prevents an increase in Zn^2+^, lysosomal dysfunction, and neuronal cell death induced by H_2_O_2_. Interestingly, the inhibition of such Zn^2+^ signaling significantly attenuates ROS-induced neuronal death. Thus, these data show the significant role of TRPM2 in the intracellular Zn^2+^ homeostasis, lysosomal, and mitochondrial functions in ROS-induced neuronal death [30].

TRPM2-mediated Ca^2+^ currents can be detected in cultured microglia. During ischemic stroke, oxidative stress induces microglial cell activation and neuroinflammation. TRPM2 produces diverse pro-inflammatory mediators such as TNF-α and interleukin-6 (IL-6) in cultured human microglial cells under buthionine–sulfoximine (BSO)-induced oxidative stress. Additionally, the excessive Ca^2+^ generated by TRPM2 was sufficient to activate mitogen-activated protein kinases (MAPK), p38, extracellular signal-regulated kinase ERK, JNK, and downstream NF-κB. However, BSO-induced increases in Ca^2+^ and the activation of MAPK and NF-κB signaling pathways were profoundly suppressed using TRPM2 inhibitors [30].

#### 3.1.4. Mitochondrial Involvement in Cerebral Ischemic Stroke

Mitochondria are critical elements of cell fate. They can promote cell survival by generating ATP to support cell activity or participate in apoptosis, leading to cell death. As very dynamic intracellular organelles, mitochondria play an essential role in the pathophysiology of ischemic neuronal death. Mitochondrial morphology is vital in the support of mitochondrial dynamics, which change from rod to small fragments after cerebral ischemia/reperfusion lesions. Fragmented mitochondria can lead to energy disturbances, oxidative stress, and the overproduction of ROS, which can induce the opening of mitochondrial transition pores. These pathological changes subsequently lead to the impairment of mitochondrial membrane potential and release of pro-apoptotic factors, such as cytochrome c (Cyt C) and apoptosis-inducing factor [31].

The structure and number of mitochondrial cristae also play a vital role in maintaining mitochondrial function. A tight cristae junction can prevent the spread of small molecules and membrane proteins under physiological conditions. However, stress-induced cristae modulation, characterized by wide cristae junctions and low cristae numbers, both result in the release of Cyt c.

The loss of ATP after ischemic injury to brain cells disrupts ionic gradients along the membranes, regulated by Na^+^/K^+^ATPase. These conditions, followed by a reversal of amino acid transporters and increases in extracellular K^+^, trigger an increase in free cytosolic Ca^2+^, and an increase in free cytosolic Ca^2+^ concentration may overload the mitochondrial proton circuit, favoring glutamate excitotoxicity [32]. In addition, increased intracellular calcium activates membrane phospholipases and protein kinases that accumulate free fatty acids and arachidonic acids. As a result, FFA, Co-A acyl, and long-chain acylcarnitines accumulate and damage mitochondrial enzyme systems [8].

Mitochondrial autophagy, called mitophagy, controls the number and quality of mitochondria, eliminating the excessive or dysfunctional mitochondria that can produce ROS-reactive oxygen species and cause cell death. During this process, mitochondria are recognized for selective autophagy. Through autophagy machines or the ubiquitin-proteasome system (UPS), mitochondrial quality control has become one of the most attractive targets for therapeutic intervention. The threshold at which mitophagy assumes a protective role in stroke by controlling mitochondrial quality requires analysis.

Transcriptional regulatory genes of mitochondrial biosynthesis mainly include coactivator 1α (PGC-1α) of the proliferative activated peroxisome proliferative receptor (PGC-1α), a potent primary regulator of mitochondrial biogenesis, nuclear respiratory factor 1 (Nrf-1), which is responsible for regulating the nuclear-encoded mitochondrial genes involved in the electron transport chain and activating the expression of mitochondrial transcription factor A, which is required to initiate mtDNA transcription and duplication.

Among the many interconnected mechanisms of endogenous defense against stroke injury, the regulation of mitochondrial bioenergetics seems to be an encouraging line of research. The convergent mitochondria event that induces cell death and tissue infarction after ischemic stroke is mainly the decrease in ATP due to lack of nutrients and oxygen. During the reperfusion phase, the supply of oxygen and nutrients induces several cellular lesions due to mitochondrial Ca^2+^ overload, mitochondrial permeability transition pore opening (MPTP), and overproduction of reactive oxygen species (ROS).

#### 3.1.5. Signaling Pathways Involved in Cerebral Ischemia

##### ROS Signaling

At low levels, ROS play an important role in normal physiological signaling and metabolic pathways. In addition, ROS can regulate some major signaling pathways for apoptosis and necrosis.

ROS increases the mitochondrial membrane’s permeability by activating p53, which reacts with cyclophilin D and opens mitochondrial permeability transition pore, leading to mitochondrial swelling. Additionally, ROS leads to the release of cytochrome c by forming an inhibitory complex during the reaction of p53 and Bcl-2 (*B-cell lymphoma 2*) family proteins, such as *Bcl-2-associated X protein* (Bax) and BH3-interacting domain death agonist (Bid), a member of the Bcl-2 family of proteins that regulate the permeabilization of the outer mitochondrial membrane [33].

##### HIF-1 Signaling

*Hypoxia-inducible factor-1α*) (HIF-1α) regulates the expression of glucose transporter 3 (GLUT3), the primary transporter of glucose in neurons. The complexity of the relationship between ROS and HIF-1α in cerebral ischemia is primarily due to the various ROS species generated by the mitochondrial O_2_ metabolism and their properties, such as the reactivity, chemical nature, specificity, and half-life of their biological targets [8].

##### CK2 Signaling

*Casein kinase 2* (CK2), a prominent oncogenic kinase, plays a key role in ROS production in cerebral ischemia. It acts as a neuroprotector by inhibiting *Nicotinamide Adenine Dinucleotide Phosphate* (NADPH) oxidase (NOX) by regulating *Recombination Protein1* (Rec1). Observation from focal ischemia in rats showed that inhibition of CK2 in the ischemic region causes PARP- 1 accumulation, resulting in the release of cytochrome C and *Apoptosis-inducing Factor*) (AIF) from the mitochondria, which activates a series of apoptotic events. In addition, recent reports show that CK2 directly phosphorylates JAK and STAT3 for *superoxide dismutase 2* (SOD2) transcription, detoxifying ROS. CK2 activates HIF-1α and NF-kB by phosphorylation to produce *vascular endothelial growth factor* (VEGF) and other angiogenic proteins, which are needed for angiogenesis and to escape hypoxia conditions [8].

##### EGFR Signaling

Several reports describe hyperactivation of the *epidermal growth factor receptor* (EGFR) pathway in both ischemic and reperfusion conditions, in which the transactivation of EGFR by phosphorylation leads to the activation of AKT—protein kinase B (PKB). AKT plays an important role in neuroprotection by regulating *forkhead transcription factor* (FKHR) and *glycogen synthase kinase 3 beta* (GSK3β), BAD or caspase-9 or other apoptogenic components, to avoid apoptosis [8]. It was also discovered that AKT, activated by ischemic ROS, provides the hypothermic conditions necessary to decrease ROS production [8].

##### TGF-β Signaling

TGF-β is highly expressed in cerebral ischemia. It modulates the expression of apoptotic (Bad) and anti-apoptotic (Bcl-2, Bcl-x1) proteins. It also crosstalks with the MAPK pathway by the transactivation of ERK½ for neuroprotection. TGF-β1 saves neurons from excitotoxicity by up-regulation of the type-1 plasminogen activator inhibitor (PAI-1) to nullify the t-PA-potentiated, NMDA-induced neuronal death in ischemia [8].

##### NF-kB Signaling

The NF-kB signaling pathway plays an essential role in cerebral ischemia. ROS-activated NF-kB regulates Bim and Noxa (77), TNF (78), IL-1, IL-6, iNOS, ICAM-1 and MMP9, cytosolic phospholipase A2, COX- 2, and microsomal prostaglandin E synthase 1. It can also directly regulate HIF-1α. All these downstream factors may cause apoptosis, cellular damage or promote cell survival in a context-dependent manner [8].

##### Calcium Signaling

Generally, Ca^2+^ signals are needed for cell-to-cell communication and survival by playing the role of secondary messenger, but Ca^2+^ overload in the ischemic area leads to calpain-mediated inhibition of the sodium–calcium exchanger (NCX) caspase and Ca^2+^ dependent endonucleases activation, which leads to apoptosis. It also helps in nitric oxide (NO) production via the calmodulin–NOS pathway. Excessive NO can combine with other ROS to produce peroxynitrite (ONOO−), which causes damage to DNA, proteins, and membrane lipids [8].

##### ACh and Related Receptors

ACh upregulation contributes to NO’s release from endothelial cells, and NO over-expression leads to vasodilation [8].

##### Nitric Oxide, RNS, and Nitrosative Stress

RNS mainly causes nitrosative stress, including two major species, NO and ONOO^−^, which participate in the process of ischemia–reperfusion injury. The basal NO concentration, which is less than 10 nmol/L, plays a key role in maintaining the normal neurocrine, immunological and vascular physiology. Excess NO can lead to the destruction of the blood–brain barrier (BBB), cell death, and inflammation. NO inhibits cytochrome c oxidase in the respiratory chain of mitochondria during ischemia. NO can react with proteins resulting in nitrosothiol formation or protein nitrosylation. NO enhances the activity of cyclooxygenase-2 (COX-2), which may mediate the excitotoxicity of glutamate, produce more ROS, and participate in the inflammatory reaction due to the pro-inflammatory prostaglandin E2, the product of the COX-2 reaction. ONOO^−^ reacts with tyrosine to form 3-nitrotyrosine, which leads to the dysfunction of some essential proteins due to changes in their structure, such as the inhibition of enzymatic activity, disruption of the cytoskeletal protein, and impaired signal transduction. ONOO^−^ reacts with tyrosine in two ways: ONOO^−^ reacts with metal ions to produce nitronium ions, which further react with tyrosine residues, and tyrosine reacts with the product of the reaction between ONOO^−^ and CO_2_. In addition, ONOO^−^ can interact with critical DNA elements, such as guanine nucleotides and sugar-phosphate backbone, causing DNA damage due to strong ONOO^−^ nitration and then activating the PARP pathway [33].

##### MAPK Signal Pathway

It has been reported that MAPK pathways can induce neuronal cell death in the cortex and hippocampus in a transient forebrain ischemia mouse model [33]. The MAPK pathway has three primary members: c-Jun NH2-terminal kinase (JNK), ERK½, and p38 MAPK. JNK and p38 MAPK play critical roles in promoting apoptosis, although the function of ERK½ in cell death is controversial [33]. The JNK and p38 MAPK pathways can be activated by apoptosis signal-regulating kinase 1 (ASK1), triggered by ROS, leading to apoptosis during ischemia-reperfusion [33].

##### Dopamine System

The dopamine (DA) system plays a pleiotropic role in stroke recovery mechanisms. While DA may not be the driving force behind the reversal of diaschisis, it is implicated in plasticity processes. Activating the cyclic adenosine monophosphate (cAMP)-response element-binding (CREB) protein plays an essential role in post-stroke recovery from motor deficits, which are of functional relevance for the position of DA in recovery from stroke as they can regulate CREB activity. Additionally, DA appears to play a role in the sprouting of spared neurons and supporting newborn neurons through the potential modulation of growth factor expression. DA is also implicated in the modulation of immune and inflammatory processes and may promote angiogenesis [34].

#### 3.1.6. Oxidative Stress Role in Cerebral Ischemic Stroke

Oxidative stress, arising from the uncontrolled production of ROS beyond the neutralizing capacity of the various endogenous defense systems, including enzymatic and non-enzymatic matters, leads to cerebral cell apoptosis and neuronal damage [35].

Neurons are generally exposed to a basic level of oxidative stress from both endogenous and exogenous sources, as are all cells in the organism. In stroke-induced brain injury, free radicals include superoxide anion radical, hydroxyl radical, and nitric oxide (NO). The damaging effects of free radicals are usually prevented or reduced by antioxidant enzymes and free radical scavengers. The main source of oxygen-derived free radicals (often called reactive oxygen species (ROS)) during ischemic stroke is the mitochondria, which produce superoxide anion radicals during the electron transport process. Another potentially critical source of superoxide in post-ischemic neurons is the metabolism of arachidonic acid through the Hipoxygenase and cyclooxygenase pathways [5].

Increasing data suggest that oxidative stress and apoptosis are closely linked phenomena in the pathophysiology of ischemic stroke. Oxygen-free radicals can also be generated by activated microglia and infiltrating peripheral leukocytes via the NADPH oxidase system following the reperfusion of ischemic tissue. This oxidation causes further tissue injury and is a critical trigger for apoptosis after ischemic stroke [5].

Due to the brain’s complex structure, relatively low antioxidant capacity, high oxidative metabolism activity, insufficient cell repair activity of neurons, and high *polyunsaturated fatty acid* (PUFA) content [36], the brain is highly susceptible to oxidative-stress-induced damage. In addition, excessive production of ROS leads to oxidative damage, including DNA harm, lipid peroxidation, and protein oxidation, which can lead to cell apoptosis or necrosis [36].

#### 3.1.7. Excitotoxicity and Apoptosis/Necrosis

Excitotoxicity is an important theory that explains the pathophysiology of cerebral ischemia [37]. The accumulation of excitotoxic amino acids (such as glutamate) [38] is the basis of excitotoxicity. A significant portion of ischemia-induced neuronal impairment is mediated by the excessive accumulation of excitatory amino acids, leading to toxic increases in intracellular calcium. Although this is an innate defensive [39] response to protect against ischemia by activating a reaction to severe cellular stress, paradoxically, this increase in intracellular calcium activates multiple signaling pathways, eventually leading to cell death. In addition, shortly after cerebral blood flow is reduced or stopped, energy-dependent cell pumps [40] fail because of the decreased glucose-dependent ATP generation, resulting in the flow of many ionic species inside the cell. This generates cellular swelling through cellular depolarization [20] and osmosis [41].

Glutamate accumulates in the extracellular space following ischemia and activates its receptors. Glutamate receptor activation provokes alterations in the concentration of intracellular ions, most notably Ca^2+^ and Na^+^ [42]. Enhancements of intracellular Na^+^ can be detrimental to neuronal cell survival earlier after ischemia [5].

#### 3.1.8. Apoptosis Molecular Mechanisms

Apoptosis is an important mechanism of secondary damage in brain tissue after cerebral ischemic/reperfusion (I/R) injury [43]. It is characterized by cell shrinkage, chromatin condensation, DNA degradation and fragmentation, and cell division into apoptotic bodies, followed by phagocytosis and degradation. Apoptosis is the primary mechanism behind the appearance of DNA in circulation [44]. Apoptosis may concur with a significant proportion of neuronal death following acute brain ischemia, leading to stroke. On the other hand, apoptosis results in intense brain injury, including cell death and loss of neurological functions [45].

Intrinsic and extrinsic signals can induce apoptosis through two major pathways: either mitochondrial (intrinsic) or death-receptor-mediated (extrinsic). Intrinsic apoptotic cascade is associated with changes in the permeability of the external mitochondrial membrane, and ROS influences this pathway by interacting with it. The ROS-regulated truncated form of the Bid protein causes Bax/Bak oligomerization and creates mega-pores in mitochondria. [8]. Next, the apoptosome complex is formed in the cytosol by activating caspase 9 and then 3 to initiate apoptosis. Increased Fas receptor expression determines mitochondrial permeability transition with ROS release, which is the primary mechanism of apoptosis induction. Bcl2 blocks NO-triggered apoptosis, foreseeing that mitochondria are the main target of apoptosis induction [8]. Mitochondrial ROS are determinants for the full activation of the caspases cascade. By disturbing the mitochondrial membrane potential, ROS generates mitochondrial pores and releases cytochrome-C. ROS can directly or indirectly raise the gating potential of pores [8]. H_2_O_2_ can induce apoptosis by initiating caspase activation and mitochondrial cytochrome C release. However, similarly to other ROS, very high doses of H2O_2_ produce necrosis in brain tissue [46]. *Schisandra Chinensis*
*Fructus* and its ingredients protect the brain by suppressing apoptosis [47].

#### 3.1.9. Autophagy

Autophagy is a cellular metabolic pathway by which damaged organelles and misfolded proteins are degraded and recycled to maintain cellular homeostasis [24]. Accumulating evidence has shown that autophagy is activated in various brain cells such as neurons, microglia, and endothelial cells during ischemic stroke, and interfering with autophagy can aggravate brain damage. [24]. Recent data have shown that several molecules required for autophagy also regulate apoptosis [45]. If the P62-dependent recruitment of necrosome to autophagy machinery is blocked, then the mechanism of cell death switches to apoptosis. Additionally, microtubule-associated protein light chain 3 (LC3) has been reported to have an apoptotic function. Hence, the established relationship between autophagy and apoptosis is closely related to LC3 and P62 signals. Oxidative stress is between the causes of autophagy [32].

#### 3.1.10. Angiogenesis

Angiogenesis is an intricate biological process regulated by various angiogenic growth factors, which are crucial in the mutual communication between endothelial cells and pericytes during the angiogenesis process [6]. VEGF is the most prominent member of the angiogenic growth factors, playing a critical role in the migration [48], proliferation [49], and tube formation of endothelial cells [50]. Previous studies have shown that Ang-1 and Ang-2 depend on VEGF to modulate angiogenesis [6]. Ang-1, as a paracrine signal [51] derived from pericyte [52], is essential for vessel maturation and stabilization [53]. Ang-2, an endogenous antagonist of Ang-1, induces vessel destabilization and participates in the early stages of vessel germination during angiogenesis [6].

It was previously shown that better collateral circulation and amplified cerebral blood flow (CBF) throughout the ischemic/hypoxic territory could provide a favorable environment for the activation of endogenous angiogenesis. At the same time, angiogenesis plays a critical role in maintaining regional cerebral blood flow in the subacute stage of cerebral ischemia. Thus, it cannot be ruled out that post-stroke angiogenesis may improve CBF in the peri-infarct brain. [6].

#### 3.1.11. Serum Proteins

Electroacupuncture (EA) may change the expression level of multiple serum proteins, including the upregulation of gelsolin, beta-2-glycoprotein I proteins, complement proteins, and downregulation of SerpinG1 protein in acute ischemic stroke patients [23]. Serum insulin and hemoglobin A1c (HbA1c) have been reduced by hyperbaric oxygen therapy (HBOT) [54].

#### 3.1.12. Brain-Derived Neurotrophic Factor

*Brain-derived neurotrophic factor* (BDNF) is the most plentiful neurotrophin in the adult brain, which possesses a remarkable ability to repair brain damage [12]. BDNF is broadly involved in synapse maturation, synaptic plasticity, the maintenance of normal cognitive function, and neurite outgrowth arborization, while BDNF dysfunction may contribute to the progression of multiple neurological diseases and psychiatric disorders. The pro-survival and neuroprotective functions of BDNF are mainly derived from two TrkB-activated signaling pathways: phosphatidylinositol 3-kinase (PI3K)/Akt pathways [55] and MAPK/ERK pathways [56]. Both play significant roles in cell cycle, division, and survival by regulating the level and activity of specific transcription factors. Recent data underline the essential role of BDNF in ischemia, suggesting its association with post-stroke mobility. It was shown that BDNF levels during the first day after stroke are notably higher among patients under 65 years compared to older patients. Additionally, low BDNF concentrations were associated with poor clinical status during 90-day follow-up. This shows that BDNF levels in the acute phase of ischemic stroke possess a prognostic value for the patient’s functional status [57]. Studies have shown that BDNF deficiency is linked to a more severe stroke pathophysiology, as BDNF plays a crucial role in developing the nervous system and promoting neuronal differentiation, cell survival, and neurogenesis. [58]. BDNF appears to be a significant candidate for the treatment of stroke [12].

#### 3.1.13. Neuroplasticity

The existing state of knowledge suggests that, due to the high plasticity of the brain [59], immediate and long-term rehabilitation allows for the neurological deficit to be reduced. In the case of stroke, neuroplasticity adaptations begin immediately after an ischemic event. After damage due to stroke, the human brain can restore its function through distributed neural networks, located in regions that were not touched by the brain infarction. These functional neural networks [60] can be found in the intact hemisphere. Changes within the ischemic penumbra [61] start relatively quickly after stroke, providing the first symptoms of early recovery. The surviving neurons after stroke [62] undergo structural and functional remodeling. Neurons compete with each other for available space in the cortex. This happens because neurons can take over the functions of nearby neurons [63].

#### 3.1.14. Blood–Brain Barrier Dysfunction

The blood–brain barrier (BBB) plays a vital role in maintaining homeostasis [52] in the neuronal microenvironment of the CNS [53]. Acute ischemic stroke enhances the interactivities of brain endothelium with extravascular CNS cells (astrocytes, microglia, neurons) and intravascular cells (platelets, leukocytes), and these interactions contribute to the injury process. The apparent result of all these responses to stroke is that cerebral vascularization involves the following phenotypes [64]: (1) poor capillary perfusion of brain tissue [65], (2) pro-adhesive for circulating cells, (3) pro-inflammatory, (4) pro-thrombogenic [5], and (5) diminished endothelial barrier function [5].

#### 3.1.15. Cell Adhesion Molecules and Ischemic Stroke

Brain ischemia triggers profound changes in cerebral microvascular endothelium. The endothelial cell activation is accompanied by the upregulation of adhesion molecules [66]. Several resident cell populations within the brain tissue can secrete pro-inflammatory mediators after an ischemic insult. These include endothelial cells, microglia, astrocytes, and neurons. The stimulation of transcription factors results in increased protein levels for cytokines and increased expression of endothelial cell adhesion molecules (CAMs) [67] in post-stroke brain tissue. Leukocytes’ infiltration of the ischemic brain region is associated with the inflammatory activation of cerebral endothelial cells, microglia/macrophages, and astrocytes. Activating these resident cell populations and immune cells stimulates the production and release of pro-inflammatory cytokines such as TNF-α and IL-1 from the ischemic tissue. In this inflammatory environment, cerebral endothelial cells increase the expression of cell surface adhesion molecules, which mediate the reinstatement of leukocytes and platelets to the ischemic region.

#### 3.1.16. Anti-Inflammation as a Therapeutic Target for Ischemic Stroke

The brain parenchyma does not elicit stereotypic immune responses. This is mainly due to the unique histological composition, as endothelial, epithelial, and glial barriers tightly regulate the accessibility of immune cells [68]. Neuronal death resulting from cerebral ischemia will induce post-stroke neuroinflammation, which can be a double-edged sword. On the one hand, cell debris can be removed, and an acute and transient immune response can facilitate repairs. On the other hand, an inflammatory response could contribute to secondary brain damage. Accumulating data demonstrate that cytokine-dependent microenvironments play significant roles in stroke [69].

Neuroinflammation was present in different types of brain injuries, including ischemic stroke. Ischemia-induced cell debris and high levels of ROS lead to neuroinflammation by activating resident microglia and astrocytes and attracting infiltrating leukocytes from circulating blood [70]. These cells grow major histocompatibility complex class II molecules and cytokines. Following the activation of microglia, the release of pro-inflammatory mediators favors the permeability of the BBB. The secretion of chemokines promotes the entry of systemic leukocytes, including neutrophils, macrophages, and lymphocytes, which share several functional features with microglia [27]. A large body of evidence implicates leukocytes in the pathogenesis of stroke injury. Neutrophils adhere to ischemic endothelial vasculature in the acute phase after stroke and infiltrate the brain parenchyma. The leukocyte recruitment may cause further damage by releasing ROS, proteases, and inflammatory mediators. The accumulation of platelets, either directly attached to endothelial cells or bound to adherent leukocytes, promotes a prothrombotic state [67].

Under inflammatory conditions, major histocompatibility complex (MHC) class II specific CD4^+^ (T helper 1/TH1) cells [71] will be stimulated. Activated CD4^+^ cells easily penetrate the BBB into the CNS following cerebral I/R. Therefore, microglia can retain and stimulate CD4+ cells that are already engaged in differentiating into T helper 1 cells, producing pro-inflammatory cytokines (IL-2, IFN-γ, TNF-α), or into T helper 2 cells, producing cytokines that underpin antibody-mediated responses (IL-4, IL-5, IL-10, lL-13). IFN-γ is thought to play an essential role in the polarization of microglia. T helper 1 cells produce pro-inflammatory cytokines such as IFN-γ, which can activate microglia into the M1 phenotype, provoke pro-inflammatory responses, and produce pro-inflammatory cytokines and oxidative metabolites [72].

IL-1β produced following the formation of an inflammasome is a critical immunoregulatory and pro-inflammatory cytokine that affects almost all cell types, such as monocytes and macrophage/microglia. In addition, after ischemic stroke, IL-1β can activate NF-kB via the activation of TLRs, enabling NF-kB to transactivate genes connected with cytokines, chemokines, and other pro-inflammatory mediators [27].

Transforming growth factor-beta (TGF-β) proteins are multifunctional cytokines with pleiotropic functions. TGF-β can modulate various biological processes, including angiogenesis, hematopoiesis, cell proliferation, migration, differentiation, and apoptosis. After ischemic stroke, TGF-β, generated by activated M2 phenotype macrophage, has an anti-inflammatory role and contributes to recovery after brain injury. TGF-β decreases microglial activation, and thus reduces the potentially harmful effects associated with stimulated microglia. In addition, TGF-β lowers the expression of other cytokines and suppresses the release of oxygen- and nitrogen-derived products. TGF-β can also activate the release of IL-1Ra and promote angiogenesis. However, its protective effects are limited to the peri-infarcted area, as TGF-β can inhibit apoptosis but not necrosis [73].

IL-4 regulates various immune and inflammatory responses, including T cell differentiation and IgE class in B cells. IL-4 is primarily produced by TH2 cells. IL-4 has a unique property, as it polarizes macrophages/microglia toward the M2 phenotype, an anti-inflammatory phenotype. Consequently, IL-4 may have a neuroprotective function, enhancing tissue repair and acting as a therapeutic factor [74].

IL-10 is an anti-inflammatory cytokine involved in immune response regulation. IL-10 can reduce inflammation after stroke. IL-10 can also reduce astrogliosis due to the direct inhibition of the pro-inflammatory cytokine TNF-α. Furthermore, IL-10 inhibits pro-inflammatory cytokines by reducing hemoglobin-induced oxidative tissue damage through the positive feedback loop with CD163, heme oxygenase-1, and IL-10 [75].

#### 3.1.17. Microglia Role in Ischemic Stroke

Microglia are small, macrophage-like glial cells representing 10–15% of the central nervous system (CNS) cells, considered the innate immune cells of the CNS. Microglia are the initial responders to tissue damage, critical modulators of the brain’s immune response. Microglia have receptors that respond to various stimuli, which may result in their activation. Microglia work closely with astrocytes to release cytokines that lead to a cascade of events that modulate the neuroinflammatory response. Meanwhile, microglia cells produce and release excitotoxic metabolites that damage the surrounding tissue. Sometimes, a short-term neuroinflammatory response can help in the recovery of damaged or infected tissue. On the contrary, a long neuroinflammatory process can affect the surrounding brain tissue. In response to different stresses, microglia are rapidly activated to differentiate into the M1 or M2 phenotype, which is involved in tissue damage and repair [76].

As frequently seen in neurobiology and neuropathology, there are many “dialectically”: complementary antagonistic relationships between cells such as microglia and astrocytes, which can be environmental determinants in peri-infarct tissue and, therefore, strongly influence the potential for neuronal plasticity, which can be both adaptive or maladaptive [77].

M1 microglia exists in a pro-inflammatory state and secretes pro-inflammatory cytokines, which have been identified to promote brain damage. Microglia M1 is activated after ischemic stroke and subsequently plays a harmful role. After experiencing ischemia/hypoxia, NF-κB is activated in the microglia and translocates from the cytoplasm to the nucleus; this triggers the release of pro-inflammatory cytokines, which cause secondary brain damage, such as interleukin (IL)-1β, IL-6, and tumor necrosis factor-α (TNF-α), in addition to producing inducible nitric oxide synthase (iNOS). Changes in M1 microglia activation may affect the prognosis of stroke. Alterations in the expression levels of M1 microglia biomarkers, including surface markers and pro-inflammatory cytokines, are common in ischemic stroke [24].

M2 microglia is anti-inflammatory and secretes anti-inflammatory cytokines and neurotrophic factors to promote brain repair. Microglia M2 is activated and reported to play a beneficial role in ischemic stroke; for example, under ischemia/hypoxia conditions, the peroxisome proliferation-activated γ receptor (PPAR γ), a transcription factor with anti-inflammatory properties, is activated and mobilized from the nucleus to the cytoplasm in microglia. Furthermore, IL-4 secreted by microglia M2 decreased the size of the infarction after ischemic stroke and improved long-term functional recovery. In addition, microglia M2-induced chitinase-3-like protein 3 (Ym1/2), IL-10, and TGF-β secretion promoted angiogenesis, thereby decreasing BBB secretion and improving stroke outcomes [24]. Additionally, M2 microglia promotes the neuronal differentiation and proliferation of neuronal stem/progenitor cells in the ipsilateral subventricular area after ischemic stroke by increasing the regulation of TGF-α expression levels; this can provide an effective therapy for neurogenesis [58]. In addition, macrophages recruited by microglia have been identified as enhancing the polarization of M2 microglia and improving stroke outcomes. [24].

M2 microglia release anti-inflammatory cytokines associated with the cAMP-response binding protein (CREB). In rats with newborn germline hemorrhage, Rh-Chemerin promoted the accumulation and proliferation of M2 microglia in the periventricular regions and suppressed the inflammatory response by the erythroid-2-factor-associated (Nrf2)-associated signaling pathways associated with the nuclear factor. Previous studies have shown that activating the M2 microglia could also promote the repair of brain tissue in other neurological diseases. [24].

Activation of the M2 microglia can be assessed by determining the expression levels of their surface markers; for example, macrophage mannose receptor 1 (CD206) and the secretion of anti-inflammatory cytokines, including IL-4, IL-10, arginase-1 (Arg-1), Ym1 and TGF-β. Changes in the percentage of activated microglia M2 type may affect the prognosis of stroke [24].

### 3.2. Non-Invasive, Non-Pharmacological Therapeutic/Rehabilitative Interventions in Acute Ischemic Stroke

Neuro-rehabilitation, encompassing physical medicine procedures including kinesiotherapy, case speech/swallowing therapy, plus different necessary rehabilitation nursing (RN) endeavors and coping strategies, are helpful to the functional and QOL improvement in stroke patients, which is possibly associated with some newer therapeutic intervention alternatives [78]. However, after neurologically and general biologically stabilizing the patient, his/her mobilization represents a primary target and a significant medical challenge, especially in the first 24 h after stroke debut, which represents a natural trend to regain such a normality dimension, but sometimes involves serious risk of related complications. Therefore, further data are needed. However, as a general principle, mobilization must be very carefully progressive, especially in the still-acute stage (days since the supra-acute event) [3]. Therefore, considering the general items described above, the main interventions will be synthetically presented. As they are both quite diverse/eclectic and ”-at least some of them- are either mixed, overlapping between different categories of such interventions, and/or taxonomically difficult/debatable on being very precisely arranged in classes”, we have chosen [77] to do this in alphabetical order.

#### 3.2.1. Acupuncture/Electro-Acupuncture (EA)

As an essential component of traditional Chinese Medicine (TCM), acupuncture has been freely clinically used for more than 2000 years. As applied through inserted—not implanted—needles, acupuncture and EA are minimally/non-invasive procedures, reported to have potential benefits for acute (including ischemic) stroke patients [79].

It is shown in the literature that acupuncture at Baihui and Shenting acupoints can significantly improve impaired functions, such as motor control and cognition, in post-stroke conditions [80].

At an intimate level, acupuncture can upregulate the expression of anti-apoptosis genes, and inhibit apoptotic signaling cascade via enhancing Akt, Bcl-2, Bcl-xL, and cIAP1/2 [81] while reducing apoptotic mediators (i.e., death receptor-5 and caspases-3, -8, and -9), with additional anti-neuroinflammatory effects (including by inducing vagal activity and the related cholinergic pathways) and promoting astrocytes and neuronal progenitor cell proliferation via Wnt/β-catenin- and ERK½-mediated pathways, especially in the focal cerebral cortex and hippocampus, due to brain-derived neurotrophic factors (NTFs)/vascular endothelial growth factor (VEGF)-mediated neurogenesis [70]. In addition, there are also asserted acupuncture abilities to augment post-ischemia CBF by promoting VEGF and angiogenin-1-mediated angiogenesis [82] and the improved release of vasoactive mediators (i.e., acetylcholine and NO); moreover, it can improve regional energy metabolism, regulate blood lipid metabolism to resist cerebral free radical damage, reduce the number of excitatory amino acids to lower neurogenic toxicity, and reduce calcium overloading [70].

EA is reported to improve motor function, mainly in the upper limbs. At an intimate level, the results found in the literature revealed that glucose metabolism significantly changed in the premotor cortex, primary motor area, bilaterally in the superior parietal lobule, and in the supplementary motor area on the unaffected hemisphere, immediately after the first EA treatment. The therapeutic intervention pathways seem to be similar to those asserted for acupuncture: increase cerebral blood flow, regulate oxidative stress, attenuate glutamate excitotoxicity, suppress Ca^2+^ overload in the infarct region of the brain, causing neuroinflammatory reactions, maintain BBB integrity, inhibit apoptosis, increase growth factor production, and induce cerebral ischemic tolerance. EA can achieve this kind of intervention (in animal ischemic models) via the inhibition of NF-kB-mediated neuronal cell apoptosis, reducing autophagy, modulating p38, ERK½, and JNK, and downregulating the expression of Nogo-A and NgR in the hippocampus of the cerebral ischemia side, thus reducing cerebral infarct volumes with improvements in neurological deficit motor and cognitive scores [80].

There is further involvement at the gene-molecular level: MiRNAs are a critical class of ncRNAs, composed of 18–24 nucleotides. EA may alter the level of miRNA expression associated with cell proliferation, including rno-miR-6216, rno-miR-206-3p, rno-miR-494-3p, and rno-miR-3473, which may be related to improved functional recovery and cerebral blood supply after stroke. In addition, EA can improve neurobehavioral function by regulating the expression of RhoA, GAP43, and PirB [23].

Regarding its physiatric component, the literature reveals considerations regarding different EA stimulation frequencies: 1–20 Hz would be involved in the suppression of autophagy within EA treatment against ischemic injury; 2 Hz (also 5–20 HZ) combined with melatonin may exert a neuroprotective effect against apoptosis by elevating B-cell lymphoma 2 (Bcl-2) and decreasing Bcl2-associated X protein(Bax), suppressing the production of Bax and reducing the malonaldehyde level, mRNA level, phosphorylated- CREB at the protein level, and the activity of glutathione peroxidase and superoxide dismutase in the hippocampus in animal models. Further, EA stimulation at a 2 Hz frequency of 1 mA intensity for 20 min may significantly expand the release of acetylcholine (ACh) from the cholinergic nerve in the ischemic brain cortex. ACh upregulation contributes to NO release from endothelial cells, and NO over-expression leads to vasodilation [23].

The same frequency interval (1–20 Hz) may reduce the levels of pro-inflammatory cytokines, including IL-1b, IL-6, and TNF-α, in rats following ischemic stroke. Furthermore, the intimate therapeutic pathways found in experimental rat models, including those with acute ischemic stroke, have been revealed to upregulate the levels of Stat5a, Stat5b, and Stat6, and various EA treatments may notably downregulate the levels of Stat1 and Stat2. In addition, EA may lower the peak expression level of heat shock protein 70 (Hsp70) and adrenocorticotrophic hormone (ACTH) to prompt neuronal repair, decrease inflammatory response and suppress excessive stress. It also may promote the gene and protein expression levels of cyclin E, cyclinD1 CDK2, and CDK4, shortening the G1-phase, omitting the G0/S and/or G1/S transition point and resulting in continued proliferation of the expression level of Robo 1 and Slit 2, which might be involved in the EA treatment mechanisms for diminishing brain infarction in the clinic, and may change the expression level of many serum proteins, including the upregulation of beta-2-glycoprotein I proteins, gelsolin, C3, C4B, and complement component I, and reduction in SerpinG1 protein in acute ischemic stroke patients [23].

Additionally, at 15 Hz, 1 mA EA stimulation at the Shuigoumay acupoint improves acute cerebral infarcted by relieving arterial spasm in rats with acute cerebral infarction by upregulating PKC activity and immunoactivity in the smooth vascular muscle of the middle cerebral artery [23].

At higher frequencies (100 Hz 2 mA), EA seems to increase NGFs levels in the BBB permeability, including the possible enhancement of aquaporin-4 expression—and possibly also with protective actions on the BBB—in the ischemic cerebral tissue [23].

Overall, acupuncture and EA stimulation, through interesting possible beneficial interferences with the vast (maybe overestimated) array of intimate and tissue lesional pathways, warrants further related studies, which might provide new therapeutic targets and optimized strategies for selecting a proper dosage for acupoint treatment [23].

#### 3.2.2. Hyperbaric Oxygen Therapy

Hyperbaric oxygen (HBO) [83] refers to “the administration of 100% oxygen at two to three times the atmospheric pressure” [7]. HBO is a ”therapeutic strategy” [7] aiming to raise the arterial oxygen tension and, thus, the oxygen supply to the tissues [84] via an increased plasma concentration in dissolved oxygen, ultimately driving the improved efficiency of cellular respiration and sustaining ATP synthesis [84], including in ischemic/hypoxic territories [52]. Progressively, HBO would ameliorate cerebral circulation, decrease brain edema, block inflammatory cascades [85] and neutrophil adhesion [86], alter the synthesis of cytokines by monocytes and increase the synthesis of heat shock proteins [87], and finally reduce the infarct size by mitigating cell death and restoring mitochondrial oxidative phosphorylation [7].

At the geno-molecular level, more than 100 downstream genes that are puzzling in terms of the glucose metabolism (glycolysis pathway and glucose transporters), cell proliferation (TGF-β3, EGF, and erythropoietin), migration, and angiogenesis (vascular endothelial growth factor) are reported in the literature. By increasing the ROS levels, HBO might also upregulate HIF-1 [8] expression. In addition, erythropoietin could exert potent neuroprotective effects [7].

In this context, cell survival would be favored by HBO, which consequently showed anti-apoptotic abilities via the mitochondrial pathway, with similar findings to those of *survivor activating factor enhancement* (SAFE) [88] stimulation: “reduced cytochrome C levels, lowered caspase-3, and caspase-9 activity, and increased Bax and Bcl-2 proteins” [7], “suppresses the p38 MAPK (involved in cell differentiation, autophagy, and apoptosis, conferring the same protection as a p38 inhibitor” and, on the p44/42 MAPK, HSP70 (an anti-apoptotic protein), increased tolerance to ischemic neuronal damage via the induction of HSP-72 synthesis [7].

Another interesting issue relates to NO dynamics after HBO treatment; HBO stimulates the endogenous production of NO, reducing neutrophil sequestration and adhesion, and improving vascular flow. However, at the same time, HBO stimulates the mRNA of both eNOS and neuronal NOS to increase the NO levels [89]. These were favorable to ischemic tolerance but were associated with increased sensitivity to convulsions and seizures during subsequent oxygen exposures, probably through the increased substrate for peroxynitrite formation [7]. HBO also exhibits antibacterial effects with bacteriostatic or bactericidal activities [90] and unclear influences on vascular tone (opposing versus inducing vasoconstriction) [7].

Consequently, HBO could promote systemic effects: it reduces the demand for hemoglobin, the cerebral edema, increases blood oxygenation and oxygen delivery to tissues [91], and contributes to the maintenance of blood–brain barrier (BBB) integrity, thus reducing the brain, diminishing the brain infarction volume, and improving the outcomes of the interaction between damaged mechanisms and related brain tissue resilience, as well as clinical, functional status. However, in humans, this is still not convincing enough, and is even controversial. However, actions that are possibly beneficial at an intimate level, as presented in the literature, could reflect a proper preconditioning mechanism (especially dose-dependent, repeated HBO type), obtained in experimental animal models, which require confirmation and translation into consequent improvements in clinical practice [7].

However, according to evidence-based medicine (EBM), related randomized, double-blind controlled trials—without excluding some possible favorable results, which may also be associated with thrombolysis within the same therapeutic window of from 3 to 6 h in acute stroke—do not provide enough evidence-based conclusions to support these beneficial outcomes. In the most recent meta-analysis, the majority, 7 of 11 randomized trials, showed no significant differences in mortality rates at 6 months in the HBO-treated patients compared with controls [7].

Additionally, in the literature, it is unclear whether HBO preconditioning protection is distinctive to the oxygen preload or does not differ from previously proven types of of preconditioning. Further, it is interesting to more clearly determine which of the two main action mechanisms of HBO, i.e., hyperoxia or hyperbaric, induce tolerance against I/R injury [7]. Hyperoxia seems to be the acting component, including through interesting negative feedback: by stimulating ROS formation, it triggers signaling pathways that reactively upregulate antioxidant enzymatic systems (catalase and superoxide dismutase—with the related combat of lipid peroxidation) with consequent protective actions against I/R-induced lesions: specifically, ”HBO increases intracellular ROS formation, “which activates both MEK1/2 and p38 MAPK. The activation of p38 MAPK initiates the transcription of the HSP32 gene [92]. Simultaneously, the activation of MEK1/2 inhibits Bach1 disassociation from small microphage-activating factor proteins, which prevents the surge of HSP32 gene transcription” [7].

#### 3.2.3. Systemic Hypothermia (Selective, Local Brain) Cooling

A preliminary specification is necessary. Regarding the therapeutical approach in acute ischemic stroke, there are two main methodological types of cooling/hypothermia interventions: (systemic) hypothermia and local (brain) cooling [93].

The average body core temperature is near 37 °C in humans [94], while hypothermia is defined as a body core temperature below 35 °C. Hypothermia can be considered a medical emergency if the body temperature falls below 32 °C, which results in multiple organ failure and even death (at hypothermia levels between 35 °C and the critical value of 32 °C, there are potentially severe consequent conditions such as blood hypercoagulability, with thrombogenic risk and immune suppression, which is prone to infection) [95]. The concept of cooling therapy comes from hibernation in some mammalian species, which results in a lowered body temperature and slowed metabolism. A hypothermic state—even a tiny, clinically feasible decline in body temperature—was found to prevent neuron death following global ischemia over 25 years ago. This finding has since prompted an intense investigation into hypothermia as a neuroprotective approach.

Although the neuroprotective mechanisms of hypothermia have yet to be entirely determined, its several consequences seem highly relevant to decreasing the severity of ischemic brain damage, including by reducing its tissue metabolism. This conceptually matches the van’t Hoff’s law regarding the relationships between milieu temperature, the histological and the intensity information of a chemical, metabolic inner reactions, the reduction in oxygen-based free-radical production, decreased excitatory neurotransmitter release, and the prevention of ATP post-ischemic loss [96], by reducing its utilization and the oxygen tension thresholds required to sustain tissue viability. Furthermore, to emphasize van’t Hoff’s law [97]: each 10 °C drop in brain temperature (37–27 °C) decreases the cerebral metabolic rate of O_2_ by 6–7% and, at least partially, the delay of the pathophysiological progress. This is also based on anti-inflammatory actions, the prevention of BBB disruption (including a reduction in MMP proteolytic activity and degradation of various components of the extracellular matrix that comprise the BBB—mainly targeting components such as vascular basement membrane proteins, agrin and laminin, and inhibiting detrimental processes such as pericyte migration), and the gene-molecular level: interference with gene expression alteration, including the inactivation of nuclear factor κB (NF-κB), a major transcription factor that modulates the expression of many inflammation-related genes under ischemic conditions, as well as the regulation of cell death and survival pathways [94].

Therefore, hypothermia in the brain decreases metabolic rate [61] but also reduces blood flow. The low basal temperature reduces cerebral oxygen consumption and glucose metabolism, maintaining valuable high-energy phosphate compounds such as ATP and the potential of hydrogen (pH) in ischemic tissues, thus minimizing the downstream effects of lactate overload and the development of acidosis [98].

Thereby, the correlation between lowering the temperature and restoring energy capacity suggests that the positive results of hypothermia at least partly concern mitochondrial functions [94].

As briefly asserted above, hypothermia may improve neurological outcomes by inhibiting various pathological aspects of the immune response following brain ischemia and injury. For example, it suppresses neutrophil extravasation and microglia activation in the affected area and reduces the level of inflammatory mediators, including oxidative stress and adhesion molecules [61].

Additional studies found in the literature report that pro-inflammatory cytokine IL-1β levels were notably reduced by local brain cooling, which simultaneously decreased the damage inflicted through vasogenic edema [99] At the same time, the anti-inflammatory cytokines, such as IL-10, are also reduced by hypothermia, suggesting that they may have a more complex role in cytokine modulation. These observations suggest that (local brain) hypothermia may also protect the BBB. Although more investigation is required, the suppression of inflammatory and downstream responses is a neuroprotective mechanism of hypothermia therapy in acute cerebral ischemia [61].

More precisely, hypothermia affects cell death processes in acute ischemic stroke models [94]. Cell death pathways can be inhibited on several levels, including modulation of the expression of Bcl-2 (B-cell lymphoma-2) family members, the release of cytochrome c, the activation of caspases, or the FasL signaling pathway. In addition, hypothermia blocked ischemia-induced translocation of protein kinase C (PKCδ) to the mitochondria and nucleus; the inhibition of PKCδ translocation to the mitochondria precludes the formation of reactive oxygen species and initiation of apoptosis. Finally, hypothermia can increase the expression of HSP70 under hypothermic conditions [87] and may alter inflammatory gene expression, serving as an essential global mediator of the dialectic-antagonistic balance between cell death and survival pathways [8].

Concerning these two methodological approaches, it should specified that (systemic) hypothermia can be ranked as mild (34–320 C), moderate (31–280 C), deep (27–110 C), and profound (<100 C). More recently (2015), the term ultra-mild hypothermia (UMH) was introduced for temperatures >350 C and 360 C, based on evidence that 360 C versus 370 C induced a bona fide intracellular cold shock response in cultured primary neurons in vitro, improved biochemical markers of brain damage after hypoxic/ischemic injury in adult rats, and confers a benefit after cardiac arrest, an effect that may not simply represent the prevention of fever [99].

One barrier to clinical unity is the rapidity of hypothermia induction; the cooling rate may be an essential determinant of clinical utility in stroke. The rapid induction of hypothermia [46] and reduction in systemic symptoms make selective cooling an attractive option. In addition, “elective cooling may provide a novel treatment that can circumvent many of the widespread objections to the use of hypothermia after stroke” [61].

Early surface cooling [17] methods include convective air blankets, water mattresses, alcohol bathing, and ice packing [100]. However, maintenance of the target temperature using these methods is difficult to control, mainly due to skin vasoconstriction and the subsequent redistribution of blood flow. However, surface cooling has the considerable advantage of pre-hospital feasibility [17]. Localized surface cooling using a cooling helmet is another method of quickly and selectively inducing hypothermic conditions without systemic complications.

Therefore, cooling using ice packs may more rapidly induce therapeutic hypothermia (TH), but does not offer reliable temperature control during the cooling or rewarming phases [101].

A methodological mix of the two main previously mentioned types seems to be the intranasal administration of cooled air; this appears to be relatively safe and has been demonstrated to concomitantly reduce core, brain, and tympanic membrane temperatures by an average of ≥1 °C. However, monitoring and controlling temperature during the cooling and rewarming process becomes complicated when utilizing an external cooling of the head and neck. Therefore, the localized surface cooling also has limited effectiveness. However, including such a mixed (systemic and local, surface) cooling might not avoid or overcome both the methodological and possible inconvenience of the invasive kind [99].

#### 3.2.4. Photobiomodulation (PBM)

Different preclinical studies have extensively studied the processes by which light energy can induce complex biological changes in cells and even systemically cause reactions that result in acute ischemic stroke in a neuroprotective response, with the neurogenesis of an inflammation reduction. However, the action mechanisms underlying the interaction between light energy and tissues are complex and not completely understood, mainly due to the different intimate chromophores that are involved: water, oxyhemoglobin, deoxyhemoglobin, myoglobin, melanin, cytochromes, flavins, chromatin. The main components of the chromosomes have the following essential constituent histones: DNA macromolecules plus nucleoproteins and possible mixtures of polyacidic polymers [102]. Most of the related studies refer to the infrared spectral wavelength (λ) within the light: electromagnetic energy. More precisely, the infrared spectrum is divided by the λ in near-infrared (NIR) (λ 760–1.400 nm), medium infrared (λ 1.400–3.000 nm), and the far infrared, especially λ between 3000 and 15,000 nM (or even up to 1 million nm = 1 mm) [103].

Previous studies have suggested that NIR acts at an intimate level, enhancing the mitochondria activity by increasing cytochrome c oxidase activity. Two NIR wavelengths partially inhibit cytochrome c oxidase activity in vitro and have the ability to interrupt the mechanisms that are accountable for cell death caused by reperfusion injury. The suppression of cytochrome c oxidase activity with 750 nm (borderline with the visible light spectrum) and/or 950 nm NIR reduces mitochondrial respiratory function, and thus prevents ROS generation in stress states, including during reperfusion. This provides robust neuroprotection [14].

The positive effects of light therapy [14] at the cellular level seem to be explicit and non-speculative but somewhat ambivalent within a complex palette of actions. For example, red/NIR light may stimulate the mitochondrial respiratory chain’s cytochrome c oxidase activity, thus increasing ATP production. In addition, this mechanism modulates the activation of transcription factors and signaling mediators such as NF-kB, resulting in long-lasting effects on cells, mainly due to the production of ROS and release of Ca^2+^ as versatile second messengers. Other positive implications of light-based therapy, aside from the stimulation of cell survival, include enhanced dendrite growth through the up-regulation of brain-derived neurotrophic factors, mediated by the activation of the extracellular, signal-regulated, kinase/cAMP-responsive, element-binding signaling pathway; the promotion of axonal protection via nerve growth-factor-induced neurite outgrowth; and neurite outgrowth and synaptogenesis stimulation promoted by MAPK activation.

The non-invasive delivery of light from an external source to the head and then into the brain is denominated transcranial PBM [77]. To implement light-based therapies in vivo, it is crucial to know depth of light penetration through the scalp, skull, and brain tissues. Therefore, different ex vivo (animals and humans) and in vivo (animals) reports, as well as simulations (Monte Carlo method [104]), have been conducted to evaluate the total light energy that can reach the damaged tissue inside the brain. However, there is no consensus on the actual light penetration into brain tissues, as it is dependent on the optical parameters, tissue characteristics, and protocol used (wavelength, irradiated area, and dose related-parameters: irradiance, exposure time, power, power density, administration rate and light source, which is possibly coherent for laser beams). The most significant limitation of this light-based technique lies in the impossibility of delivering adequate, non-invasive light energy into the damaged area inside the patient’s brain, which is able to induce the corresponding biological effect. In this regard, light irradiation via the nasal or oral cavity demonstrated beneficial effects on Alzheimer’s disease, depression, and anxiety. These non-invasive techniques may be an alternative to delivering light energy into the brain, but they would not suit all stroke subtypes due to the human brain structure [105].

Literature reports regarding preclinical studies show that PBM treatment for ischemic stroke involves the use of light from the visible to NIR portions of the spectrum (wavelength ~600–1100 nm) with a relatively low fluence or energy density (1–30 J/cm^2^) to avoid thermal effects or burns in the tissue. On the other hand, penetration through the scalp and skull of beams within physical spectral lambda raises questions regarding their penetrability to the brain. The main related methodological paradigms that are considered are the time that light-based treatment is started (usually within the first 24 h of stroke onset), the exposure time and several repetitions (generally less than 60 min, with more than one irradiation session), light operation mode (continuous or pulsed), coherence (for low-level lasers) and convergence (for light-emitting diodes (LED)) [77,105].

Considering all the above, and despite variability in the protocols implemented in animal models of ischemic stroke, many results seem to indicate that PBM therapy (at an average transmission of ~1–20% with a penetration length of ~3–30 mm) applied <24 h from stroke onset reduces the neurological damage evaluated by functional tests, final infarct volume assessment, and mortality rate or cell neurogenesis. However, most in vivo PBM studies for stroke are limited by the size and light penetration capacity across different brain tissues. LED-PBM treatment, initiated 24 h after stroke, does not achieve significant recovery results when analyzed using lesion reduction, behavioral tests, and functional resonance imaging as objectification modalities [105].

In the literature, three connected clinical PMB trials were conducted on acute ischemic stroke (within 24 h of onset). These trials were named ”Neuro Thera Effectiveness and Safety Trial (NEST)” [17]. These showed good preliminary results (NEST-1, which demonstrated the safety and effectiveness of transcranial low-level laser therapy under these conditions), NEST-2 (ClinicalTrials.gov Identifier: NCT00419705). A total of 660 patients between 40–90 years of age failed to prove efficacy; therefore, the futility data obtained in more than ½ of the recruited cases were withdrawn (”Phase III Clinical End Point Device Trial”)****. NEST-3 (ClinicalTrials.gov Identifier: NCT01120301) enrolled the largest amount (1000) of patients compared to the two previous such trials, but still did not produce satisfactory outcomes. Furthermore, a neuroprotective effect was not found when considering the 90-day functional outcome of the patients [105].

#### 3.2.5. Therapeutic Gases

Medical or therapeutic gases [106] are another non-drug approach with possible neuroprotective beneficial effects against acute ischemic stroke. These gases range from traditional gases (oxygen and carbon dioxide) to nitric oxide, carbon monoxide, xenon, argon, helium, and hydrogen sulfide [107].

*Oxygen:* Hypoxia is a critical component contributing to neuronal death in stroke. Increasing brain tissue oxygenation has long been considered a logical strategy against cerebral ischemia [8]. Oxygen, a commonly used therapeutic agent, has distinct advantages over pharmaceutical drugs. The ease of diffusion across the BBB and the relatively safe, well-tolerated, and minimal, dose-limiting side-effects position oxygen therapy as likely to be effective at improving oxygen supply to the brain tissue. Maintaining tissue oxygenation is critical for neural cells and is severely affected by stroke. The blockage of blood flow during a cerebral ischemic event induced a rapid decrease in localized interstitial pO_2_, as measured by paramagnetic resonance oximetry (30% of pre-ischemic values in the penumbra and 4% in the core during cerebral ischemia). Oxygen treatment improves cerebral perfusion and oxygenation in ischemic regions, as well as inhibiting peri-infarct depolarizations [17].

In stroke patients treated with inhalation oxygen at a flow rate of 45 L/min through a face mask for 8 h, lactate, a marker for aerobic metabolism, decreased during oxygen administration. The decrease in lactate suggests that the aerobic metabolism can be improved following stroke by normal baric oxygen (NBO). Minimizing energy metabolic dysfunction and restoring tissue oxygenation in the acute phase of stroke (i.e., en route to a hospital) might prolong the available time-window for reperfusion therapies and the subsequent rescue of the penumbral region. gp91phox, a significant source of ROS, is the catalytic subunit of NADPH oxidase. Providing a possible mechanism of how NBO may reduce oxidative stress, NBO inhibited the upregulation of gp91phox. In addition, the genomic deletion of gp91phox led to a situation where NBO could no longer reduce BBB leakage following cerebral ischemia. The effect of NBO on nitric oxide generation was also examined following stroke. NBO administered during cerebral ischemia delayed and attenuated early nitric oxide generation, possibly inhibiting nNOS [106].

NBO treatment inhibited the induction of MMP-9 in the ischemic brain and was associated with reduced occludin degradation, Evan’s blue extravasation, and hemispheric swelling. These all suggest that NBO treatment protects against BBB lesions. Furthermore, the mechanism by which NBO suppresses MMP-9 and attenuates BBB damage appears to involve the aforementioned inhibition of NADPH oxidase, as the inhibition of NADPH oxidase by apocynin or gp91phox knockout resulted in much lower amplitudes in MMP-9 induction and BBB leakage in ischemia [106].

NBO has been shown to induce vasoconstriction, improve dynamic functional cerebral blood flow in salvageable tissue, increase O_2_ availability, reduce metabolic disturbances, and protect the blood–brain barrier. In addition, it is simple to administer, non-invasive, inexpensive, widely available, and can be started promptly after stroke onset.

A larger-scale trial was imitated in 2007 at Massachusetts General Hospital to compare the safety and therapeutic efficacy of NBO (started within 9 h of symptom onset) to that of room air (30–45 L/min for 8 h). Unfortunately, after 85 enrollments, this was terminated because of the imbalance in deaths favoring the control arm (ClinicalTrials.gov, Identifier: NCT00414726). Finally, in 2010, a randomized pilot study in India evaluated the effect of normobaric high-flow oxygen therapy in acute ischemic stroke by the clinical (NIHSS, mRS) and radiological (MRI) measurements. Twenty patients were randomized to receive NBO. In contrast with the above studies, the result showed that NBO did not improve the clinical scores of stroke outcomes in patients with acute ischemic stroke. Furthermore, adverse effects were observed, including cardiac/pulmonary side effects and free radical injury [17].

In addition to oxygen, laboratory and clinical evidence has also found that other gases may play a beneficial role in treating stroke. For example, xenon, argon, and helium administration demonstrate effective neuroprotection [17].

Emerging data support the hypothesis that *hydrogen*, which has the advantage of crossing the BBB quickly, might act as a free radical scavenger of hydroxyl radicals, thus promoting neuroprotection. Hydrogen (2%) reduced the number of toxic hydroxyl radicals and significantly reduced infarct volume following transient MCAO in rats. Studies using other rat models confirmed the effects of hydrogen on the improvement in neurological function. A recent clinical study treated 13 cerebral ischemia patients with hydrogen inhalation and showed that 3–4% hydrogen inhalation over 30 min was feasible and safe in stroke patients [17].

*Helium* was proposed as a therapeutic gas in 1934, and, subsequently, heliox (helium/oxygen gas mixture) was presented as a potential therapy for upper- and lower-airway obstruction. In a side-by-side comparison of the efficacy of hyperoxia and that of heliox in a transient MCAO rat model, heliox was more effective than hyperoxia in reducing infarct volume and improving neurological deficits if initiated immediately after the onset of ischemia. The most protective outcome was observed when heliox (70% helium) was administered directly after occlusion. Additionally, the defensive actions of heliox therapy were demonstrated in traumatic brain injury, cardiac ischemia, and hypoxia-ischemia, but the exact mechanisms behind these protective effects are still unclear [17].

*Xenon* is used as an anesthetic gas and, because of its N-methyl-D-aspartate (NMDA) receptor inhibition properties, has potential neuroprotective roles during brain ischemia. Xenon has also been noted to have synergistic/antagonistic neuroprotective effects when used with other therapeutic strategies. Xenon can attenuate rtPA-mediated protection by acting as an inhibitor of rtPA. Thus, any clinical use of xenon would have to occur either in the absence of or following rtPA administration [17].

*Argon:* Studies have supported the neuroprotective effects of *argon* both in vitro and in vivo, and it carries the advantage of being more affordable and common than xenon while still able to protect against excitotoxic injury. In addition, argon is thought to induce GABA neurotransmitters and thus inhibit NMDA, but, as GABA receptor activation has additional downstream pathway influences, the protective mechanisms at work during argon administration have yet to be fully determined. Still, promising preclinical results for argon following ischemia suggest that it could be further examined as a potential neuroprotectant in ischemic stroke [17].

*Carbon dioxide* and oxygen are vital vasodilators in CBF self-regulation, as this inner homeostatic primary mechanism is often impaired in patients with acute ischemic stroke and persistent proximal arterial steno-occlusive disorders. Carbon dioxide (CO_2_) is a standard gas in the air, which has been widely used in physiatric balneary treatment. CO_2_ is a low-fat, soluble molecular gas with a strong diffusion capacity, which can cross the blood–brain barrier. In healthy people, CO_2_ is maintained in a narrow homeostatic serum titer (35–45 mmHg) by specific physiological mechanisms. PaCO_2_ is the balance between CO_2_ production and disposal. The possible neuroprotective effect of CO_2_ is worth discussing. CO_2_ is a vasodilator, and its low concentration in the blood is thought to cause cerebral vasoconstriction. Mild and moderate increases in PaCO_2_ 60–100 mmHg levels have neuroprotective effects against cerebral ischemia/reperfusion injury caused by the adaptive above-mentioned vascular reactive vascular, and this protective effect may be related to its beneficial influence on apoptosis-regulating proteins. The circulatory response to hypercapnia increases the percentage of CBF in the human brain, with an approximately 6% per 1 mmHg rise in the arterial tension of CO_2_ (PaCO_2_), an essential mediator of CO_2_-related vessel dilatation [106].

*Hydrogen sulfide (H_2_S)*: Accumulating shreds of evidence indicate that H_2_S plays an important role in stroke. The H_2_S neuroprotective effect is dose-dependent. Only when its level is relatively low can H_2_S yield neuroprotection, while a high dose may lead to neurotoxicity. H_2_S is also an essential biochemical component within the mud, and sulfurous waters, used in balneotherapy. It induces a wide range of physiological responses such as blood pressure modulation, neuromodulation in the brain and within the vasculature, protective against ischemic reperfusion injury, and anti-inflammatory reactions. H_2_S can penetrate the skin and mucosae and act at the geno-molecular level [107].

#### 3.2.6. Transcranial Direct Currents Stimulations (tDCS)

It should be emphasized that most of the works in the literature discuss the effects of transcranial direct current stimulation (tDCS) interventions on chronic post-stroke patients. Papers referring to interventions in the acute stage of ischemic stroke are scarce. Additionally, a meta-analysis of human studies reported that tDCS had a significant effect in patients with a chronic stroke as compared to acute and subacute stroke [108].

Brain stimulation is a propitious area of research as it allows the excitability of the target area to be directly manipulated, as the brain belongs to the central nervous system, which contains the most considerable amount of (specific) preformatted excitable tissue structures [109]. Some methods, such as tDCS, aim to restore the interhemispheric balance by inhibiting the healthy hemisphere or stimulating the lesioned one. Additional details are presented elsewhere, including polarity-related effects [77]. Here, we only mention two studies: one relatively recent study [110] and another very new study [111], reporting the somewhat conflicting results of tDCS interventions, i.e., which electrode (Anodal [110] vs. Cathodal [111]) exerts beneficial effects on stroke patients.

The latter study asserts that cathodal tDCS significantly ”improved the level of neurological deficit and the brain morphology, reduced the brain damage area and apoptotic index, and increased the number of Nissl body in MCAO (middle cerebral artery occlusion -o.n.) rats”. Additionally, cathodal tDCS inhibited the ”high level of NSE (Neuron-specific Enolase—o.n.), inflammatory factors, Caspase 3 and Bax/Bcl2 ratio” and also ”the activation of astrocyte and microglia” in the respective experimental animal model. Moreover, a very recent paper presents the results of a pilot randomized control trial concerning “Cathodal Transcranial Direct Current Stimulation intervention in Acute Ischemic Stroke” [112]. Applied in ”acute human MCA (Meddle Cerebral Artery—o.n.) territory stroke”, “No significant difference was found between the active and sham groups…suggest potential benefits of C-tDCS” in patients with acute ischemic stroke [111].

These methods noninvasively modulate brain activity, induce brain plasticity, and facilitate post-stroke recovery [63]. Various studies have suggested that electrical stimulation affects oligodendrocytes or Schwann cells, supporting neural conduction in the nervous system. However, it is still unclear how the association between oligodendrocytes and neurons will be influenced when both cell types are simultaneously stimulated. It is conceivable that external stimuli might change the contents of soluble factors released from both oligodendrocytes and neurons, activating numerous signaling pathways. Therefore, the stimuli parameters, such as the duration, current input, and frequency, need to be optimized for each disease condition [113]. With a clearer understanding of molecular mechanisms, the modulation of neural activity could become a novel therapeutic strategy for the treatment of demyelinating diseases [114]. Regarding the clinical expression of stimulatory effects of tDCS, a recent study [108] determined that, in acute ischemic stroke, tDCS seems to be an ”effective adjuvant to conventional rehabilitation techniques,” providing significant improvements ”in all functional motor outcomes and somatosensory functions” [115].

#### 3.2.7. Transcranial Magnetic Stimulation

Transcranial magnetic stimulation (TMS), a non-invasive brain stimulation (NIBS) or neuromodulation technique consisting of alternating magnetic fields being applied to the head has been proposed to enhance cortical excitability in the affected hemisphere [56]. More recently, repetitive TMS (rTMS), which uses repeated, short magnetic bursts, has been found to exert longer-lasting effects than non-repetitive TMS and is currently being investigated for the treatment of several neurological and psychiatric conditions, including stroke. Some related details in this domain are presented elsewhere [77].

A topical experimental study revealed that 60 Hz TMS treatment counteracts oxidative stress and cell death via activation of the antioxidant system after experimental autoimmune encephalomyelitis in rats, with the following functional expression: attenuation of tail and limb paralysis. In addition, it was revealed that high-frequency rTMS increases cell proliferation and synaptic plasticity and inhibits apoptotic cell death through the activation of BDNF, CREB, ERK, and AKT signaling pathways in neuronal cells after oxygen-glucose deprivation in vitro. Furthermore, it was shown that rTMS ameliorates cognitive deficits, lesion size via the activation of Bcl-2, and inhibition of Bax and cytochrome-c release and inflammation after ischemic stroke post-hemi-cerebellectomy in rats. Additionally, in acute pathophysiological events and subacute recovery processes after occlusion of the middle cerebral artery (MCA) in mice, high-frequency rTMS (20 Hz) is reported to induce neuronal survival and regional cerebral blood flow (CBF) while reducing infarct volume and apoptotic cell death [116]. However, although rTMS has gained significant interest for the treatment of patients with neurodegenerative disorders, there are inconsistent data on the efficacy of high- and low-frequency rTMS from cellular and molecular perspectives (despite the intimate actions discussed above) [117].

However, the imbalance in interhemispheric functional connectivity includes an increase in the excitability of the contralesional hemisphere, which has been described in both basic research and clinical studies. This hyper-excitability was explained by the recruitment of circuits that are typically involved in other functions by unmasking the uncrossed fibers to the paretic arm/leg and reducing transcallosal inhibitory pathways [38].

### 3.3. Clinical Studies/Trials on Non-Invasive, Non-Pharmacological Therapeutic/Rehabilitative Interventions in Acute Ischemic Stroke

Clinical studies/trials for non-invasive, non-pharmacological therapeutic/rehabilitative interventions in acute ischemic stroke were searched on https://clinicaltrials.gov, https://trialsearch.who.int/ (accessed on 16 December 2021) and https://www.clinicaltrialsregister.eu/ (accessed on 16 December 2021) for clinical trials, using the following search items, identified during synthetic review: electro-/acupuncture, hyperbaric oxygen therapy, hypothermia/cooling, photobiomodulation, therapeutic gases, transcranial direct current stimulation, or transcranial magnetic stimulation. In addition, the inclusion criteria were fixed regarding the characteristics of the intervention: non-invasive and non-pharmacological, patients with acute ischemic stroke (as diagnosed using any recognized diagnostic criteria), aged from 18 to elderly, all genders. Exclusion criteria correspond to patients aged less than 18 years. All the literature and clinical trial databases were searched in November and December 2021, and the interrogations were carried out for the period: 01.01.2017–01.12.2021.

#### 3.3.1. Presentation of Clinical Studies/Trials on Non-Invasive, Non-Pharmacological Therapeutic/Rehabilitative Interventions in Acute Ischemic Stroke

##### Acupuncture

Clinical trials for Electro-/Acupuncture interventions in acute ischemic stroke were searched on https://clinicaltrials.gov (accessed on 16 December 2021): https://clinicaltrials.gov/ct2/show/NCT03097055 (accessed on 16 December 2021); https://trialsearch.who.int (accessed on 16 December 2021): 1—already included: ANAIS; and https://www.clinicaltrialsregister.eu (accessed on 16 December 2021): 0 (Figure 4).

##### Hyperbaric Oxygen Therapy

Clinical trials for Hyperbaric Oxygen Therapy interventions in acute ischemic stroke were searched on https://clinicaltrials.gov (accessed on 16 December 2021): https://clinicaltrials.gov/ct2/show/NCT03431402 (accessed on 16 December 2021); https://trialsearch.who.int (accessed on 16 December 2021): ChiCTR2000040026, ChiCTR2000035074; and https://www.clinicaltrialsregister.eu (accessed on 16 December 2021): 0 (Figure 5).

##### Hypothermia/Cooling

Clinical trials for Hypothermia/Cooling interventions in acute ischemic stroke were searched on https://clinicaltrials.gov (accessed on 16 December 2021): https://clinicaltrials.gov/ct2/show/NCT04554797 (accessed on 16 December 2021), https://clinicaltrials.gov/ct2/show/NCT04695236 (accessed on 16 December 2021), https://clinicaltrials.gov/ct2/show/NCT03804060 (accessed on 16 December 2021) (Figure 6), https://trialsearch.who.int (accessed on 16 December 2021): 4 (Figure 7); and https://www.clinicaltrialsregister.eu (accessed on 16 December 2021): 0.

##### Photobiomodulation

Clinical trials for Photobiomodulation interventions in acute ischemic stroke were searched on https://clinicaltrials.gov (accessed on 16 December 2021):, https://trialsearch.who.int (accessed on 16 December 2021): 0; and https://www.clinicaltrialsregister.eu (accessed on 16 December 2021): 0.

##### Therapeutic Gases

Clinical trials for Therapeutic Gases interventions in acute ischemic stroke were searched on https://clinicaltrials.gov (accessed on 16 December 2021): https://clinicaltrials.gov/ct2/show/NCT04839224 (accessed on 16 December 2021) (Figure 8), https://trialsearch.who.int (accessed on 16 December 2021): 0; and https://www.clinicaltrialsregister.eu (accessed on 16 December 2021): 0.

##### Transcranial Direct Currents Stimulations

Clinical trials for Therapeutic Gases interventions in acute ischemic stroke were searched on https://clinicaltrials.gov (accessed on 16 December 2021):, https://clinicaltrials.gov/ct2/show/NCT04061577 (accessed on 16 December 2021), https://clinicaltrials.gov/ct2/show/NCT04938076 (accessed on 16 December 2021), https://clinicaltrials.gov/ct2/show/NCT04287231 (accessed on 16 December 2021), https://clinicaltrials.gov/ct2/show/NCT04687033 (accessed on 16 December 2021), https://clinicaltrials.gov/ct2/show/NCT03297450 (accessed on 16 December 2021) (Figure 9); https://trialsearch.who.int (accessed on 16 December 2021): 0; and https://www.clinicaltrialsregister.eu (accessed on 16 December 2021): 0.

##### Transcranial Magnetic Stimulation

Clinical trials for Transcranial Magnetic Stimulation interventions in acute ischemic stroke were searched on https://clinicaltrials.gov (accessed on 16 December 2021): 0, https://trialsearch.who.int (accessed on 16 December 2021): 0; and https://www.clinicaltrialsregister.eu (accessed on 16 December 2021): 0.

#### 3.3.2. Meta-Analysis of the Data Obtained from Identified Clinical Studies

As the final step, we performed a related meta-analysis. Fourteen control trials were included in this step, with 838 human subjects in total. The smallest sample size was 8, and the largest was 172 (Table 1). The meta-analysis investigated the frequency of non-invasive non-pharmacological interventions in acute ischemic stroke. Our forest plot, also known as a blobbogram, shows a graphical display of the estimated results from several scientific studies addressing the same question and the overall results. Uneven trials only look at different concepts [118]. Statistical heterogeneity is only apparent after the analysis of the results. Heterogeneity may be judged graphically (by looking at the forest plot) and statistically measured. In a forest plot from the systematic review, the error bars for each trial include the summary result, which suggests that statistical heterogeneity is not a problem and the message is consistent (Figure 10). When determining whether significant heterogeneity exists, a high *p*-value for the χ2 test of hetero¬geneity is good news because it suggests that the heterogeneity is insignificant, and that one can continue to summarise the results.

Binary Random-EffectsModel, Metric: Proportion

EstimateLower boundUpper boundStd. error*p*-Value0.0700.0480.0920.011<0.001Heterogeneity: tau^2^: 0.002; Q(df = 13): 782.360; Het. *p*-Value: <0.001; I^2^: 98.338.

As observed in the meta-analysis, the relevant minimum number of patients is around 60. Therefore, the non-invasive, non-pharmacological therapeutic/rehabilitative interventions found in the literature with the most extensive overall number of patients were hypothermia/cooling, with 268 (31.94%), and tDCS, with 260 (31.11%) patients included in clinical trials.

## 4. Discussion

Stroke is divided into three clinical phases based on the pathological characteristics and timing post-stroke: acute, subacute, and chronic. The duration and pathological severity of the three phases vary among individuals. The variations depend on the specific conditions of individuals regarding the location and size of the lesion, the rapidity of arterial occlusion, the presence of cerebrovascular collateral circulation, the metabolic state of brain tissue, patient’s age and medical comorbidities. Generally, the acute phase of stroke is the first 48 h after the onset of stroke symptoms; the subacute phase of stroke is the period between 48 h and 6 weeks, to 3 or 6 months post-stroke, whereas the chronic phase starts from 3 to 6 months after stroke [119].

In translational research, animal stroke models represent a fundamental tool for the following: (a) to understand the molecular mechanisms underlying the short- and long-term physiological responses of all individual neuronal systems, and the whole brain, to injury; (b) to set up new therapeutic strategies to salvage and rescue those structures; (c) to find the best temporal intervention window with pharmacological and rehabilitation interventions. The complexity of all cascade events notwithstanding, the choice of a reliable model is a research priority to reconcile the existing marked differences between rodents and humans regarding both the cerebral vasculature and the nervous system architecture [120].

The current therapies were mainly aimed at the four cornerstones of acute ischemic stroke (AIS): (a) the prevention and treatment of secondary complications; (b) reperfusion strategies directed at arterial recanalization; (c) neuroprotective strategies aimed at the cellular and metabolic targets; (d) the inhibition or modulation of the inflammatory response. The detailed underlying mechanisms of this injury remain unknown; however, many researchers have come to emphasize the role of inflammation, autophagy, and apoptosis as essential contributors [121]. In addition, spontaneous neural repair also occurs after stroke and continues for many weeks, possibly years, for some behaviors, especially language and knowledge. Thus, a better understanding of automatic repair provides helpful information for new synergistic therapeutic strategies. This point is emphasized by the fact that the treatments that promote repair are often offered in the context of spontaneous repair.

Global cerebral ischemia (GCI) commonly occurs in cardiac arrest (CA) patients. The primary determinant of the clinical outcome of CA is a hypoxic brain injury. Therefore, GCI is not only a major direct cause of death but is also associated with a significant neurologic disability after CA. Clinically, GCI is distinct from focal cerebral ischemia (FCI), which is often associated with atherosclerotic or embolic stroke. Initial neuroprotective management is essential in GCI, and the therapeutic induction of hypothermia is known to improve neurologic recovery after severe cerebral ischemia. Unfortunately, despite rigorous research and control, the GCI outcomes associated with CA have not appreciably changed [122].

The efficacy of the current, non-invasive, non-pharmacological, therapeutic/rehabilitative interventions in acute ischemic stroke must be objectified/quantified. For this purpose, it might be useful in clinical practice. These interventions include clinical–functional evaluation instruments (such as the Glasgow Coma Scale (GCS), Glasgow Outcome Score Scale (GOS), (modified) Rankin Scale, National Institutes of Health Stroke Scale (NIHSS) etc.) [100], as well as neuroimaging and/or neurophysiological investigations (structural and functional magnetic resonance imaging, positron emission tomography (PET), single-photon emission computed tomography (SPECT), and repetitive/transcranial magnetic stimulation (r/TMS)), and, depending on the general biological and neuro-functional condition of the patient, functional near-infrared spectroscopy (f/NIR) [5], and possibly different immuno-(cyto)/histochemical assays.

Cardiac abnormality, such as myocardial ischemia, is associated with stroke patients. The most common ECG fiducial changes include depressed ST-segments, prolonged QT-interval, flat or inverted T-waves, and U-waves. As ischemic stroke impairs autonomic function, a predominance of sympathetic activity is installed. This is the reason why ECG heart rate variability features are used as clinical biomarkers to understand the changes in the autonomic nervous system (ANS) after stroke [123].

Electrocardiography (ECG) [124], electroencephalography (EEG) [125], electromyography (EMG) [126] and others, including the above-mentioned, signal-based, non-invasive assessment instruments [127], are also prospective predictive methods for the exhaustive prognostics of acute stroke. These entail a spontaneous biological and neurological evolution, post-stroke rehabilitation management efficacy, and related general and specific outcomes. More recently, machine learning approaches to monitoring large ECG datasets enable early stroke prognostics and post-stroke recovery using the cardiac activity profile.

This review did not settle on an approach to the monitoring/diagnostic and/or prognostic evaluative methods, as our focus, as emphasized in the title, was on the non-invasive, non-pharmacological therapeutical/rehabilitative interventions in acute ischemic stroke. Although this might be a limitation of our paper, we had, to consider the enormous amount of data and consequent difficulty following the results in a readable paper, as well as the necessity of framing the data within a reasonable editorial space.

## 5. Conclusions

This systematic review synthesizes the current findings on acute ischemic stroke interventions, which are described as non-invasive and non-pharmacological. To develop new and effective therapies for acute ischemic stroke, it is also necessary to investigate the geno-molecular, cellular, tissue and systemic mechanisms underlying brain damage and neuroprotection. The non-invasive, non-pharmacological therapeutic/rehabilitative interventions for acute ischemic stroke are mainly holistic endeavors; most of them require validation before they are regularly used in practice. Despite a significant amount of research aiming to find decisive, curative, non-invasive, non-pharmaceutical therapeutical/rehabilitative interventions, including in acute ischemic stroke, as well as pharmacological ones (except for early thrombolytic—but these also have significant limitations), the translation of some encouraging preclinical outcomes from ”bench side to bedside” [77] has yet to appear.

## Figures and Tables

**Figure 1 ijms-23-00907-f001:**
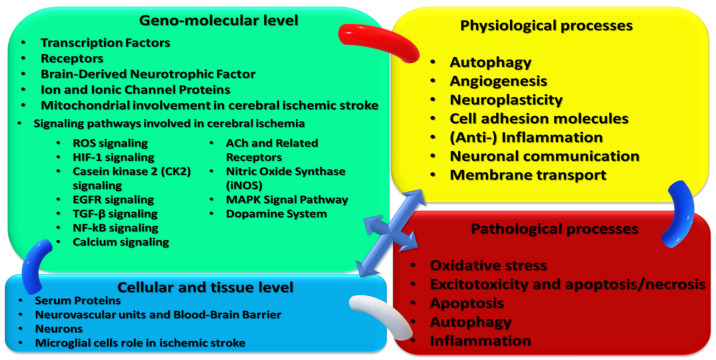
Integrative synoptic diagram of basic intimate structures and physiological/pathological processes targeted by non-invasive, non-pharmacological therapeutic/rehabilitative interventions in acute Ischemic stroke.

**Figure 2 ijms-23-00907-f002:**
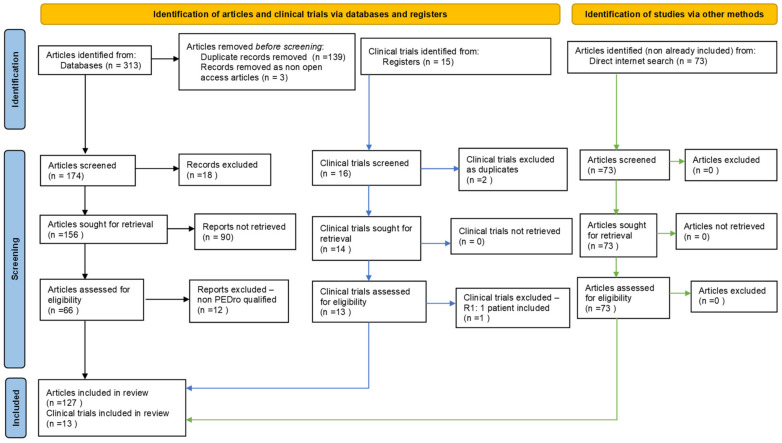
Our adapted PRISMA-type flow diagram [21].

**Figure 3 ijms-23-00907-f003:**
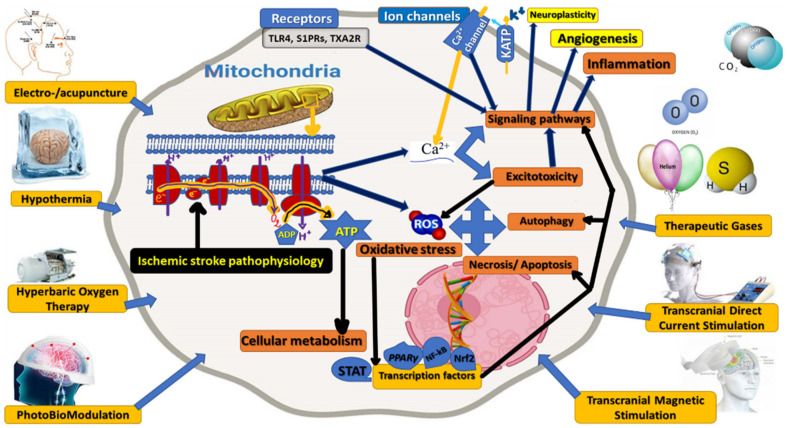
Cellular and molecular mechanisms as presumptive therapeutic/rehabilitative targets in acute ischemic stroke for non-invasive, non-pharmacological therapeutic/rehabilitative interventions: electro-/acupuncture, hyperbaric oxygen therapy, hypothermia/cooling, photobiomodulation, therapeutic gases, transcranial direct currents stimulations, or transcranial magnetic stimulation.

**Figure 4 ijms-23-00907-f004:**
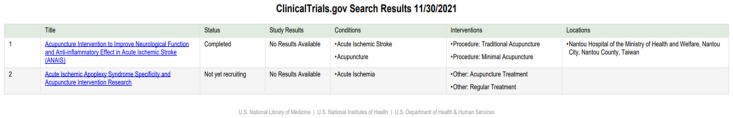
Clinical trials for Electro-/Acupuncture interventions in acute ischemic stroke.

**Figure 5 ijms-23-00907-f005:**
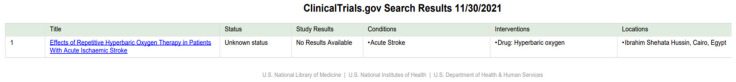
Clinical trials for Hyperbaric Oxygen Therapy interventions in acute ischemic stroke.

**Figure 6 ijms-23-00907-f006:**
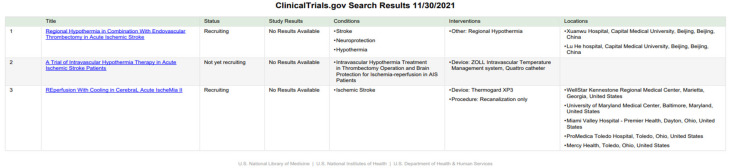
Clinical trials for Hypothermia/Cooling interventions in acute ischemic stroke on https://clinicaltrials.gov (accessed on 16 December 2021).

**Figure 7 ijms-23-00907-f007:**
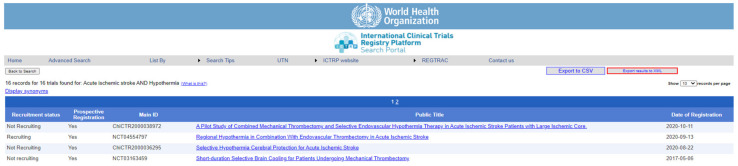
Clinical trials for Hypothermia/Cooling interventions in acute ischemic stroke on https://trialsearch.who.int (accessed on 16 December 2021).

**Figure 8 ijms-23-00907-f008:**
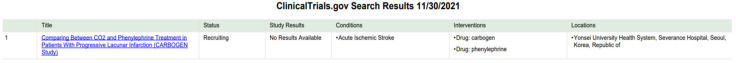
Clinical trials for Therapeutic Gases interventions in acute ischemic stroke on https://clinicaltrials.gov (accessed on 16 December 2021).

**Figure 9 ijms-23-00907-f009:**
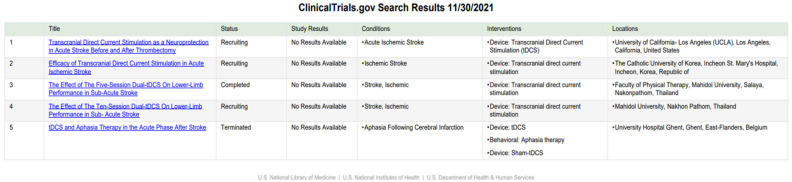
Clinical trials for Transcranial Direct Currents Stimulations interventions in acute ischemic stroke on https://clinicaltrials.gov (accessed on 16 December 2021).

**Figure 10 ijms-23-00907-f010:**
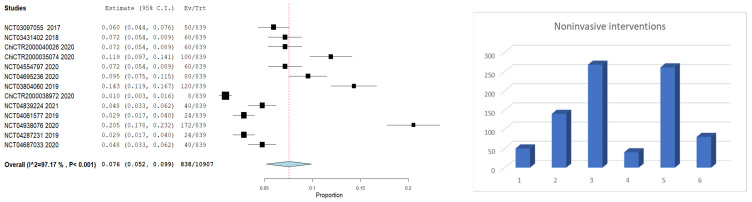
Forest plot—frequency of non-invasive, non-pharmacological interventions.

**Table 1 ijms-23-00907-t001:** The clinical trials that satisfied all the previous filtering criteria/PRISMA stages selected for qualitative synthesis were included in our meta-analysis to determine the frequency with which non-invasive, non-pharmacological interventions were used for acute ischemic stroke.

No.	Study	Start Year	END Year	N—Total Patients	Acupuncture	Hyperbaric Oxygen Therapy	Hypothermia/Cooling	Therapeutic Gases (CO_2_)	Transcranial Direct Current Stimulation (tDCS)	CONTROL
0	AXIS LEGEND			1	2	3	4	5	6	7
**1**	**NCT03097055**	**2017**	**2019**	**50**	**50**					
**2**	**NCT03431402**	**2018**	**2020**	**60**		**60**				
**3**	**ChiCTR2000040026**	**2020**	**2022**	**60**		**30**				**30**
**4**	**ChiCTR2000035074**	**2020**	**2022**	**100**		**50**				**50**
**5**	**NCT04554797**	**2020**	**2020**	**60**			**60**			
**6**	**NCT04695236**	**2020**	**2021**	**80**			**80**			
**7**	**NCT03804060**	**2019**	**2020**	**120**			**120**			
**8**	**ChiCTR2000038972**	**2020**	**2021**	**8**			**8**			
**9**	**NCT04839224**	**2021**	**2022**	**40**				**40**		
**10**	**NCT04061577**	**2019**	**2025**	**24**					**24**	
**11**	**NCT04938076**	**2020**	**2022**	**172**					**172**	
**12**	**NCT04287231**	**2019**	**2020**	**24**					**24**	
**13**	**NCT04687033**	**2020**	**2021**	**40**					**40**	
	**TOTAL**			**838**	**50**	**140**	**268**	**40**	**260**	**80**
	**%**			**100**	**5.96**	**16.69**	**31.94**	**4.77**	**31.11**	**9.54**

## Data Availability

Appendix A regarding the selection process were attached during the submission process. This systematic review is registered on the PROSPERO—International prospective register of systematic reviews—online platform, No. CRD42021293713.

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
