# Peer review of "Cellular and Molecular Targets for Non-Invasive, Non-Pharmacological Therapeutic/Rehabilitative Interventions in Acute Ischemic Stroke"

_ijms, 2022, doi:10.3390/ijms23020907_

Round 1

Reviewer 1 Report

The present work reviews the cellular and molecular mechanisms involved in acute ischemic stroke, as well as the commonly suggested non-pharmacological therapeutic strategies. Although I can appreciate the amount of effort put into this work, there are several points that should be reconsidered before it would qualify for publication. 

 As a general comment, the manuscript is not easy to follow and many paragraphs should be rephrased. I would suggest splitting long sentences into more and shorter ones, avoiding many (parentheses) and -between dashes-. Further, there are several grammatical and syntactic issues.

Specific comments: 

- Overall the abstract is rather extended; some details could be skipped given that they are meticulously presented in the main text. 

- It is very important that the authors followed PRISMA guidelines, securing the quality of included papers. However, I believe that the flow diagram alone (presented in the main text or as a supplementary file) would be enough and there is no specific need for the materials/methods part. 

- The first part of the manuscript, listing the cellular and molecular factors/pathways/mechanisms involved in stroke is chaotic. This can be partially explained by the plethora of mechanisms involved; yet, the review in its current form does not serve its purpose of "bringing this knowledge together".  Possibly, using diagrams or tables would help in this direction. 

- The authors have used reviews rather than original articles in several cases. Unfortunately, I also noticed that many sentences and some paragraphs are copied and pasted almost as a whole.

- There is a last paragraph that includes a meta-analysis of clinical trials. This is an important part of this paper and should be more highlighted. Moreover, given the very low number of clinical trials, it is essential to emphasize (preferably earlier in the manuscript) that all the findings mentioned in the main text concern animal models, in order to avoid misinterpretations.

Author Response

Dear reviewer, the authors warmly thank you for your thorough and highly professional review, which helped us improve our manuscript significantly. Regarding your comments and suggestions, please find herein below our answers, point-by-point:

  • Overall the abstract is rather extended; some details could be skipped given that they are meticulously presented in the main text. 
  • Resolved- abstract less than a half-page.
  • It is very important that the authors followed PRISMA guidelines, securing the quality of included papers. However, I believe that the flow diagram alone (presented in the main text or as a supplementary file) would be enough and there is no specific need for the materials/methods part. 
  • Resolved, meantime respecting all the required PRISMA 2020  guidelines (http://prisma-statement.org) 
  • The first part of the manuscript, listing the cellular and molecular factors/pathways/mechanisms involved in stroke is chaotic. This can be partially explained by the plethora of mechanisms involved; yet, the review in its current form does not serve its purpose of "bringing this knowledge together".  Possibly, using diagrams or tables would help in this direction. 
  • Resolved - we added a synoptic diagram.
  • .The authors have used reviews rather than original articles in several cases. Unfortunately, I also noticed that many sentences and some paragraphs are copied and pasted almost as a whole.
  • We herewith attached sen you the results of our Grammarly test for similitudes detection we have applied for this manuscript.
  • There is a last paragraph that includes a meta-analysis of clinical trials. This is an important part of this paper and should be more highlighted. Moreover, given the very low number of clinical trials, it is essential to emphasize (preferably earlier in the manuscript) that all the findings mentioned in the main text concern animal models, in order to avoid misinterpretations.
  • Resolved, starting from introduction, page 3.

Reviewer 2 Report

This study aimed to review Cellular and Molecular Targets for Non-invasive, Non-pharmacological therapeutic/ rehabilitative interventions in Acute Ischemic Stroke. I have the following major suggestions:

  • The abstract should be more concise.
  • Please add a paragraph about the contribution of this article in a bulleted form at the end part of the Introduction section.
  • Which novelty do authors claim for this review? The authors should discuss the motivations of this study.
  • Authors should add tables of literature review outcomes to increased readability.
  • The authors should provide the physiological signal-based non-invasive interventions in Acute Ischemic Stroke in the introduction section. Several ML/DL-based disease prediction systems have been already reported for stroke, acute diseases, like doi:10.3390/brainsci11070900, doi.org/10.3390/s21165334, doi.org/ 1109/ICCE46568.2020.9043098. Authors should improve the manuscript with appropriate case studies related to acute diseases.
  • Authors should add domain-specific figures of subsections to visualize the review outcome.
  • Manuscript is unnecessarily long and difficult to follow. Many concepts were repeated. Manuscript should be reorganized and should be concisely written.
  • Authors should extend the scope of sensors, state-of-art stroke patient monitoring technologies. Authors should explore state-of-art ML/DL health monitoring systems, such as a real-time patient monitoring system (ieee access 8 (2020) 213574) and a real-time ecg cardiac monitoring system (ieee access 9 (2021) 123146).
  • Figure 2 is so naive and needs to improve with more domain-specific information.
  • The discussion section needs to be improved. The authors should discuss the strength and weaknesses of studies and make directions for therapeutic/ rehabilitative interventions in Acute Ischemic Stroke in the discussion section.

Author Response

Dear reviewer, the authors warmly thank you for your thorough and highly professional review, which helped us improve our manuscript significantly. Regarding your comments and suggestions, please find herein below our answers, point-by-point:

  1. The abstract should be more concise.
  2. Please add a paragraph about the contribution of this article in a bulleted form at the end part of the Introduction section.
  3. Which novelty do authors claim for this review? The authors should discuss the motivations of this study.
  4. Authors should add tables of literature review outcomes to increased readability.
  5. The authors should provide the physiological signal-based non-invasive interventions in Acute Ischemic Stroke in the introduction section. Several ML/DL-based disease prediction systems have been already reported for stroke, acute diseases, like doi:10.3390/brainsci11070900, doi.org/10.3390/s21165334, doi.org/ 1109/ICCE46568.2020.9043098. Authors should improve the manuscript with appropriate case studies related to acute diseases.
  6. Authors should add domain-specific figures of subsections to visualize the review outcome.
  7. Manuscript is unnecessarily long and difficult to follow. Many concepts were repeated. Manuscript should be reorganized and should be concisely written.
  8. Authors should extend the scope of sensors, state-of-art stroke patient monitoring technologies. Authors should explore state-of-art ML/DL health monitoring systems, such as a real-time patient monitoring system (ieee access 8 (2020) 213574) and a real-time ecg cardiac monitoring system (ieee access 9 (2021) 123146).
  9. Figure 2 is so naive and needs to improve with more domain-specific information.
  10. The discussion section needs to be improved. The authors should discuss the strength and weaknesses of studies and make directions for therapeutic/ rehabilitative interventions in Acute Ischemic Stroke in the discussion section.

 31 Dec 2021 06:12:12

  1. Resolved
  2. Resolved
  3. Resolved
  4. Kindly please also the supplementary materials
  5. Hopefully resolved
  6. Hopefully resolved
  7. Resolved (now our manuscript is shorter with 4 pages, yet we have made some adding to the content in order to match the different suggestions of the reviewers)
  8. Resolved
  9. Resolved
  10. Resolved

Round 2

Reviewer 1 Report

The manuscript has been greatly improved.

Reviewer 2 Report

Thanks for addressing comments.